# Exploring the potential antimalarial properties, safety profile, and phytochemical composition of *Mesua ferrea* Linn.

Atthaphon Konyanee[1,2], Prapaporn Chaniad[3,4], Arisara Phuwajaroanpong[2,4], Walaiporn Plirat[3,4], Parnpen Viriyavejakul[5], Abdi Wira Septama[6], Chuchard Punsawad[2,6]*

1 College of Graduate Studies, Walailak University, Nakhon Si Thammarat, Thailand, 2 School of Allied Health Sciences, Walailak University, Nakhon Si Thammarat, Thailand, 3 School of Medicine, Walailak University, Nakhon Si Thammarat, Thailand, 4 Research Center in Tropical Pathobiology, Walailak University, Nakhon Si Thammarat, Thailand, 5 Department of Tropical Pathology, Faculty of Tropical Medicine, Mahidol University, Bangkok, Thailand, 6 Research Center for Pharmaceutical Ingredient and Traditional Medicine, National Research and Innovation Agency (BRIN), Cibinong Science Center, West Java, Indonesia

* chuchard.pu@wu.ac.th

**Data Availability Statement:** All relevant data are within the paper and its Supporting Information.

## Abstract

The increased resistance of *Plasmodium falciparum* to artemisinin and its partner drugs poses a serious challenge to global malaria control and elimination programs. This study aimed to investigate the therapeutic potential of *Mesua ferrea* Linn., a medicinal plant, as a source for novel antimalarial compounds. In this study, we conducted *in vitro* assays to evaluate the antimalarial activity and cytotoxicity of crude extracts derived from *M. ferrea* L. leaves and branches. Subsequently, the most promising extracts were subjected to assessments of their antimalarial efficacy and acute oral toxicity tests in mouse models. Furthermore, selected crude extracts underwent gas chromatography-mass spectrometry (GC-MS) analysis to identify their phytochemical compositions. Our findings revealed that the ethanolic extract of *M. ferrea* L. branches (EMFB) exhibited high antimalarial activity, with an $IC_{50}$ value of 4.54 μg/mL, closely followed by the ethanolic extract of *M. ferrea* L. leaves (EMFL), with an $IC_{50}$ value of 6.76 μg/mL. Conversely, the aqueous extracts of *M. ferrea* L. branches (AMFB) and leaves (AMFL) exhibited weak and inactive activity, respectively. The selected extracts, EMFB and EMFL, demonstrated significant dose-dependent parasitemia suppression, reaching a maximum of 62.61% and 54.48% at 600 mg/kg body weight, respectively. Furthermore, the acute oral toxicity test indicated no observable toxicity at a dosage of 2,000 mg/kg body weight for both extracts. GC-MS analysis revealed abundant compounds in the EMFB, such as oleamide, *cis*-β-farnesene, alloaromadendrene, physcion, palmitic acid, 5-hydroxymethylfurfural, and 4H-pyran-4-one, 2,3-dihydro-3,5-dihydroxy-6-methyl-, while the EMFL contained friedelin, friedelinol, betulin, β-caryophyllene, oleamide, and 5-hydroxymethylfurfural. Notably, both extracts shared several phytochemical compounds, including 4H-pyran-4-one, 2,3-dihydro-3,5-dihydroxy-6-methyl-, 5-hydroxymethylfurfural, α-copaene, cyperene, β-caryophyllene, alloaromadendrene, palmitic acid, ethyl palmitate, and oleamide. Additionally, further study is needed to isolate and

**Funding:** This project is funded by the National Research Council of Thailand (NRCT) (Contract No. N41A670189) and the Walailak University Graduate Research Fund (Contract No. CGS-RF-2023/03). The funders had no role in the study design, data collection and analysis, decision to publish, or preparation of the manuscript.

**Competing interests:** The authors declare no competing interests regarding the publication of this study.

**Abbreviations:** ACTs, Artemisinin-based combination therapies; ALP, Alkaline phosphatase; ALT, Alanine aminotransferase; AST, Aspartate aminotransferase; BUN, Blood urea nitrogen; $CC_{50}$, 50% cytotoxic concentration; DMSO, Dimethyl sulfoxide; EI, Electron ionization; GC-MS, Gas chromatography-mass spectrometry; GMS, Greater Mekong Subregion; H&E, Hematoxylin and eosin; $IC_{50}$, Half-maximal inhibitory concentration; ICR, Institute Cancer Research; $LD_{50}$, : Median lethal dose; MIC, Minimum inhibitory concentration; MTT, 3-(4,5-dimethylthiazol-2-yl)-2,5-diphenyltetrazolium bromide; NIST, National Institute of Standards and Technology; pLDH, parasite lactate dehydrogenase; RMP, Rodent malaria parasites; SI, Selectivity index.

characterize these bioactive compounds from *M. ferrea* L. leaves and branches for their potential utilization as scaffolds in the development of novel antimalarial drugs.

## Introduction

In clinical practice, artemisinin-based combination therapies (ACTs) are used as the first-line treatment for uncomplicated *Plasmodium falciparum* malaria. Over the past two decades, this treatment methodology has saved the lives of millions of people with the disease [1]. However, the development of resistance to artemisinin in *P. falciparum* was initially reported in Cambodia and subsequently spread throughout the Greater Mekong Subregion (GMS), including Thailand, Vietnam, Myanmar, and Laos [2, 3]. In addition, partial resistance to artemisinin has also been reported in African countries, including Uganda, Tanzania, Rwanda, and Ethiopia [4]. Artemisinin resistance is characterized by the delayed clearance of *P. falciparum* following treatments with artemisinin monotherapy, or ACTs [5], which has resulted in an increased exposure of parasites to partner drugs, such as piperaquine and mefloquine, leading to further drug resistance. Resistance to both drugs is reported in the GMS, and they are associated with high rates of ACT treatment failures [3, 6]. Thus, the emergence of artemisinin resistance in *P. falciparum* poses a significant challenge to malaria control and elimination efforts [7]. Consequently, the discovery of potent and novel antimalarial drugs has become an urgent healthcare issue in malaria-endemic countries to effectively control the spread of resistant parasites [8].

Medicinal plants have long been recognized as valuable sources of chemicals with therapeutic potential, and they continue to be an important source for discovering novel therapeutic leads [9]. Several plant-derived natural products have been discovered and successfully introduced into clinics for the treatment of a wide variety of human diseases. For instance, morphine is isolated from *Papaver somniferum*, paclitaxel is isolated from *Taxus brevifolia*, while vincristine is isolated from *Catharanthus roseus* [10]. Within the context of malaria treatment, antimalarial drugs have been traditionally derived from medicinal plants. Notably, quinine, isolated from the bark of *Cinchona* L. species, provided the scaffold for the synthesis of chloroquine and derivatives. Similarly, artemisinin, isolated from *Artemisia annua* L., served as the scaffold for the development of artemisinin-based derivatives [11]. In addition, lapachol, isolated from *Tabebuia avellanedae*, was used for the synthesis of atovaquone [12, 13]. The exceptional examples of antimalarial drugs derived from medicinal plants provide tremendous inspiration for the discovery of novel and potent antimalarial drugs, particularly with regard to the emergence of parasite resistance to currently available antimalarial drugs [14].

*Mesua ferrea* L. is a tropical evergreen tree belonging to the *Calophyllaceae* family and is native to various Asian countries, including Bangladesh, India, Nepal, Sri Lanka, Myanmar, and Thailand [15, 16]. *M. ferrea* L. has been extensively used to treat various diseases or conditions, including fever, cough, asthma, itchiness, nausea, dyspepsia, and renal disease [17, 18]. This plant also demonstrates several biological properties, including antimicrobial, anti-inflammatory, anticancer, antioxidant, and antivenom properties [19]. In addition, *M. ferrea* L. is included in Ayurvedic formulations such as Chyavanprash and Brahma Rasayana, which are known for boosting immunity [19, 20]. Previous studies revealed that *Mesua* coumarins isolated from *M. ferrea* L. blossoms exhibit antimalarial activity against the chloroquine-resistant *P. falciparum* W2 strain [21]. In addition, the *M. ferrea* L. stem bark subfractions exhibit anticancer properties against human colorectal carcinoma cells (HCT 116 cell line) through

triggering apoptosis and modulating different cell signaling pathways [22]. Furthermore, *M. ferrea* L. leaf extracts demonstrated strong antibacterial activity against *Staphylococcus aureus* at a low minimum inhibitory concentration (MIC) of 48.00 μg/mL [23]. Additionally, a coumarin mesuol isolated from *M. ferrea* L. seed oil demonstrated the ability to restore hematological profiles in Wistar rats with cyclophosphamide-induced myelosuppression [19].

In our prior investigation, we found that the ethanolic extract derived from *M. ferrea* L. flowers, a component of the Pra-Sa-Chan-Dang (PSCD) remedy, a Thai traditional medicine, exhibited high antimalarial activity against the chloroquine-resistant *P. falciparum* K1 strain, with an $IC_{50}$ value of 4.30 ± 0.73 μg/mL. Furthermore, the PSCD remedy also demonstrated efficacy against the *P. berghei* ANKA-infected mouse model with 78.36 ± 3.46% parasitemia suppression [24]. Previous studies have also highlighted the antioxidant properties of the ethanolic extract of *M. ferrea* L. leaves [25]. Furthermore, furanocoumarin isolated from *M. ferrea* L. leaves and branches exhibited selective inhibition of cytochrome P450 1B1 (CYP1B1), suggesting a potential role in mitigating DNA mutations and carcinogenesis [26]. However, there remains a gap in understanding the potential antimalarial activity of *M. ferrea* L. leaves and branches. Therefore, this study aimed to address this gap by providing robust scientific evidence. The primary focus of our investigation was to evaluate the *in vitro* antimalarial and cytotoxic activity of the crude extracts derived from *M. ferrea* L. leaves and branches. Subsequently, the most promising extracts undergo comprehensive antimalarial efficacy and acute oral toxicity tests in mouse models. Moreover, the selected extracts also investigate their phytochemical compositions. Through this study, we aim to contribute valuable information regarding the antimalarial efficacy and safety profile of crude extracts derived from *M. ferrea* L. leaves and branches.

## Methods

### Ethics approval and consent participate

Studies involving human participants were carried out in accordance with relevant guidelines and regulations of the Declaration of Helsinki. The protocol was approved by the Ethics Committee in Human Research Walailak University prior to participant recruitment (approval number: WUEC-23-062-01). The recruitment period for blood sample collection in this study was from April 1, 2023 to May 4, 2023. Written informed consent was obtained from participants before the blood samples were collected by an expert medical technologist. The samples were then prepared for use in the *in vitro* culture of the human malaria parasite, *P. falciparum* K1 strain. This study also received approval from the Institutional Biosafety Committee, Walailak University (approval number: WU-IBC-66-007). In addition, animal ethical clearance was obtained from the Walailak University Institutional Animal Care and Use Committee (WU-IACUC) prior to conducting the experiments (approval number: WU-ACUC-66019). The research and animal care staff were appropriately trained in the handling and use of laboratory animals. All protocols in this study were conducted in accordance with the relevant guidelines and regulations for the use of animals, in compliance with the Animal Research: Reporting of In Vivo Experiments (ARRIVE) guidelines. All surgical procedures were performed under isoflurane anesthesia, and all efforts were made to minimize the suffering of animals.

### Plant materials and extraction procedure

*M. ferrea* L. mature leaves (blue grey to dark greens) and mature branches (reddish brown) were collected at Walailak University in Nakhon Si Thammarat, Thailand, in March 2022 (GPS coordinates: 8˚38'34.8"N, 99˚53'58.9"E). The authentication of the plant specimens was

performed by Asst. Prof. Dr. Supreeya Yuenyongsawad, a specialist botanist affiliated with the School of Pharmacy at Walailak University, Thailand. The voucher herbarium specimen number of *M. ferrea* L. leaves and branches, designated as SMD122007001, has been deposited at the School of Medicine, Walailak University, Thailand. Extraction procedures were performed by following a previously established protocol [27, 28]. In brief, the plant specimens were thoroughly cleaned with distilled water to remove any undesired residues prior to drying in a hot air oven at 60°C for 72 h. The dried plant specimens were finely chopped into small pieces. Each plant specimen weighed 60 g and was extracted using two distinct methods, including maceration and decoction. For the maceration method, 95% ethanol (Merck, Darmstadt, Germany) was utilized as the solvent, involving the combination of 60 g of dried plant specimens with 600 mL of 95% ethanol. The mixture was left at room temperature for 72 h. In contrast, the decoction method employed distilled water as the solvent, with 60 g of dried plant specimens being mixed with 600 mL of distilled water and boiled at 90–100°C for 30 min. Both maceration and decoction methods involved the repetition of the extraction process twice for the remaining plant residues. The plant extracts were combined and filtered using filter paper (Whatman 1, GE Healthcare, USA). The filtered ethanolic extracts underwent solvent evaporation using a rotary vacuum evaporator (Heidolph Hei–VAP, Schwabach, Germany) at 45°C. Subsequently, the ethanolic and aqueous extracts were dried using a freeze-drying machine (Christ gamma 2–16 LSCplus, Osterode am Harz, Germany). The resulting ethanolic and aqueous extracts were stored in a refrigerator at 4°C in the sealed containers until further use.

## *P. falciparum* K1 strain cultivation

The chloroquine-resistant *P. falciparum* K1 strain was cultivated following a previously established protocol by Trager and Jensen, with some modifications [29]. Briefly, the *P. falciparum* K1 strain was cultivated on uninfected $O^+$ human erythrocytes collected from healthy participants. The cultivation was maintained in the Roswell Park Memorial Institute (RPMI) 1640 medium (Gibco, Carlsbad, CA, USA), supplemented with 2 mg/mL of sodium bicarbonate (Sigma-Aldrich, St. Louis, MO, USA), 10 µg/mL of hypoxanthine (Sigma-Aldrich, St. Louis, MO, USA), 4.8 mg/mL of HEPES (Himedia, Mumbai, India), 0.5% of AlbuMAX™ II (Gibco, Auckland, New Zealand), and 2.5 µg/mL of gentamicin (Sigma-Aldrich, New Delhi, India). The culture was then maintained at 37°C in a 5% $CO_2$ incubator, following a previously established protocol [30]. Parasitemia was assessed by preparing a thin blood film, followed by staining with Giemsa stain (Biotechnical, Bangkok, Thailand), and observation under a light microscope (Olympus CX31, Tokyo, Japan) with a 100X oil immersion objective lens.

## Evaluation of *in vitro* antimalarial activity using the parasite lactate dehydrogenase (pLDH) assay

The pLDH assay, developed by Makler with some modifications, was employed in this study to evaluate pLDH produced from live *Plasmodium* parasites [31]. The EMFL, EMFB, AMFL, and AMFB were solubilized in dimethyl sulfoxide (DMSO) (Merck, Darmstadt, Germany) to create the stock solutions, respectively. These stock solutions were then further diluted to final concentrations ranging from 0.78 to 100 µg/mL to serve as test substances in the pLDH assay. The antimalarial drugs, which include artesunate (ARS) and chloroquine (CQ) (Sigma-Aldrich, St. Louis, MO, USA), were employed as the positive control, while DMSO served as the negative control. Additionally, uninfected erythrocytes were utilized as the baseline control. Briefly, *P. falciparum* K1-infected erythrocytes (with 2% parasitemia and 2% hematocrit) were transferred to a 96-well plate, with 199 µL added per well. The test substances, positive controls, and negative controls were then introduced into each well (1 µL in triplicates of each

concentration). Subsequently, the test plates were incubated at 37˚C in a 5% $CO_2$ incubator for a duration of 72 h. Following the incubation periods, the test plates underwent three freeze–thaw cycles (–20˚C for 30 min and 37˚C for 30 min). This procedure was carried out to achieve complete lysis of erythrocytes, thereby facilitating the release of pLDH enzymes. The 20 μL aliquots of solutions obtained from lysed erythrocytes were subsequently transferred into new 96-well plates. Each well of the microplates was pre-loaded with 100 μL of Malstat reagent and 20 μL of a solution containing *p*-nitroblue tetrazolium chloride (NBT) (Merck, Darmstadt, Germany) and phenazine ethosulfate (PES) (Sigma-Aldrich, St. Louis, MO, USA). Subsequently, the test plates were incubated in the dark for 1 h. Following the incubation periods, the results were assessed by measuring the absorbance at 650 nm using a Multiskan SkyHigh microplate spectrophotometer (Thermo Fisher Scientific, USA) in accordance with the established protocol [30]. The half-maximal inhibitory concentration ($IC_{50}$) was estimated in GraphPad Prism 6 (GraphPad Software, La Jolla, CA, USA) using nonlinear regression curve analysis.

## Evaluation of *in vitro* cytotoxicity using the 3-(4,5-dimethylthiazol-2-yl)-2,5-diphenyltetrazolium bromide (MTT) assay

The MTT assay was employed for this purpose, according to an established protocol [30]. Briefly, Vero cells (ATCC number CCL-81) were cultured in Dulbecco's Modified Eagle Medium (DMEM) (Gibco, Carlsbad, CA, USA), supplemented with 10% fetal bovine serum (Sigma-Aldrich, New Delhi, India). Subsequently, the Vero cells were aliquoted and plated onto 96-well plates at a cell density of $1 \times 10^4$ cells per well in 199 μL of culture medium. The cells were then allowed to settle overnight at 37˚C in a 5% $CO_2$ incubator. Subsequently, the cultured Vero cells were treated with 1 μL of each test substance at concentrations ranging from 6.25 to 800 μg/mL in triplicate for each concentration. Doxorubicin (DX) (Sigma-Aldrich, New Delhi, India) was employed as the positive control, while DMSO served as the negative control. The test plates were then incubated at the same conditions for 48 h. Following the incubation periods, the cell culture supernatant was carefully aspirated from each well. Subsequently, 50 μL of a 5 mg/mL MTT solution (Thermo Fisher Scientific, Oregon, USA) was added to each well. The test plates were then incubated under the same conditions for an additional 2 h. After this incubation period, the supernatant was once again aspirated and replaced with 100 μL per well of DMSO to facilitate the dissolution of the formazan crystals resulting from the reaction. The colorimetric reaction was quantified by measuring the absorbance at 560 nm using a Multiskan SkyHigh microplate spectrophotometer (Thermo Fisher Scientific, USA). To ensure accuracy, the background absorbance at 670 nm was subtracted from the absorbance reading at 560 nm for each well. The 50% cytotoxic concentration ($CC_{50}$) was estimated in GraphPad Prism 6 (GraphPad Software, La Jolla, CA, USA) using nonlinear regression curve analysis.

## Selectivity index (SI)

SI values were employed to identify candidate crude extracts derived from *M. ferrea* L. leaves and branches for further evaluation in subsequent experiments of *in vivo* antimalarial activity. The SI values were calculated as the ratio between cytotoxicity against Vero cells ($CC_{50}$) and antimalarial activity against the *P. falciparum* K1 strain ($IC_{50}$) for each crude extract, as per Eq (1) below. SI values >10 indicate an optimal safety window between the effective concentration against malaria parasites and the toxic concentration in mammalian cells [32, 33]. Therefore, crude extracts with SI values >10 and exhibiting promising antimalarial activity were

selected for subsequent experiments.

$$\text{Selectivity index (SI)} = CC_{50}/IC_{50} \tag{1}$$

## Identification of phytochemical constituents by GC-MS analysis

The phytochemical constituents in the EMFL and EMFB were identified by using a triple quadrupole GC-MS/MS instrument (Agilent 7890B GC system, Agilent Technologies, Santa Clara, CA, USA) installed with an HP-5ms column (30 m × 0.25 mm, 0.25 μm), employing helium as the carrier gas with a flow rate set at 1.0 mL/min, according to an established protocol [34]. Detection was performed using a mass selective detector of the Agilent 7000C GC/MS Triple Quad (Agilent Technologies, Santa Clara, CA, USA) with an electron ionization (EI) voltage of 70 eV and an EI source temperature of 250˚C. The mass spectral scan range was 33–600 m/z. The column temperature was initially set at 60˚C for 2 min, followed by a gradual increase to 150˚C at a rate of 10˚C/min. Subsequently, it was further raised to 300˚C at a rate of 5˚C/min, maintaining this temperature for 14 min. The sample was injected at a volume of 1.0 μL in split mode, with a split ratio of 20:1. Identification of phytochemical constituents in the EMFL and EMFB was conducted by comparing the mass spectra with the compounds in the National Institute of Standards and Technology (NIST) 2011 mass spectral library. Compounds in the EMFL and EMFB with mass spectra exhibiting more than 80% similarity to those in the NIST library were selected for identification in this study.

## Mice models and rodent malaria parasites (RMP)

The ninety male Institute Cancer Research (ICR) mice, aged 6–8 weeks and weighing 25–30 g, were obtained from Nomura Siam International Co., Ltd., Bangkok, Thailand. Mice were acclimatized in a controlled laboratory environment for 7 days before the experiments began, with five mice housed per cage. The environment was maintained at a temperature of 22 ± 3˚C, with humidity ranging between 50–60%. The mice had *ad libitum* access to commercial pelleted foods and clean water. The lighting conditions in the animal room followed a 12-h light/dark cycle. In addition, the trained animal care staff works diligently to ensure the welfare of animals and proper hygiene by cleaning and removing all waste from the cages on a daily basis [35]. The RMP, *Plasmodium berghei*, strain ANKA (catalog no. MRA-311), was obtained from the Biodefense and Emerging Infectious Research Resources Repository, National Institute of Allergy and Infectious Diseases, and National Institutes of Health, USA, and contributed by Thomas F. McCutchan. For the propagation of the RMP, *P. berghei* ANKA-infected mouse blood was intraperitoneally injected into donor mice. Once the parasitemia of the donor mice reached 20–30%, anesthesia was induced using an inhalation method with 2% isoflurane (Piramal Critical Care, PA, USA) mixed with oxygen as the carrier gas within an anesthesia induction chamber. This method aims to reduce the potential distress of animals. The success of anesthesia induction in the mice was assessed by evaluating the toe-pinch pain reflex following established protocols [36, 37]. Following anesthesia, blood was drawn *via* cardiac puncture and collected in heparinized tubes. The blood was then diluted using 0.9% physiological saline, adjusting the dilution based on the parasitemia in the donor mice. Subsequently, the diluted blood was used for injection into the experimental mice for the 4-day suppressive test. The experimental durations in animal models typically ranged from 15 to 30 days. During the experiment, the condition of the animals was routinely monitored twice daily. Any animals showing signs of coma or experiencing severe symptoms, including immobility, lack of body extension, and unresponsiveness to external stimuli, were immediately euthanized in a humane manner to alleviate pain and distress [38]. Euthanasia was performed using deep isoflurane anesthesia in accordance with the established protocol [38].

## Four-day suppressive test (Peter's test)

The test was conducted following the methodology described by Peters, with some modifications [30, 39]. Briefly, 45 male ICR mice were intraperitoneally inoculated with 200 μL of $1 \times 10^7$ cells/mL of *P. berghei* ANKA-infected erythrocytes. The mice were randomly divided into nine groups, each consisting of five mice per group. Post-infection, at 4 h, each group of mice received an oral administration of different treatment substances in a volume of 200 μL *via* oral gavage. Group 1 served as the negative control group and received the vehicle solvents used for dissolving the crude extracts and antimalarial drugs (7% Tween 80 and 3% ethanol in distilled water); groups 2 and 3 served as the positive control group (antimalarial drugs) and received 25 mg/kg body weight of CQ and 6 mg/kg body weight of ARS, respectively; groups 4, 5, and 6 served as the experimental groups and received EMFL at doses of 200, 400, and 600 mg/kg body weight, respectively; and groups 7, 8, and 9 also served as the experimental groups and received EMFB at doses of 200, 400, and 600 mg/kg body weight, respectively. The dosages were selected to contain low, moderate, and high doses of crude extracts at 200, 400, and 600 mg/kg body weight, respectively, according to previous studies [30, 40, 41]. Oral administration continued for three consecutive days (24, 48, and 72 h post-infection). On day 4 (96 h post-infection), parasitemia was assessed by collecting blood from the tail vein for each mouse in all the groups. Thin blood films were prepared, stained with Giemsa stain (Biotechnical, Bangkok, Thailand), and examined under a light microscope (Olympus CX31, Tokyo, Japan) with a 100X oil immersion objective lens. The percentage of parasitemia was determined by enumerating the number of infected and non-infected erythrocytes in randomly selected fields during microscopic examinations. This counting was performed in five fields, and the percentage of parasitemia was calculated using Eq (2). Simultaneously, the percentage of suppression was calculated using Eq (3) [42]. On day 4, after concluding the experiment, all mice were euthanized using deep isoflurane anesthesia in order to minimize pain and distress.

$$\%\text{parasitemia} = [A/B] \times 100 \qquad (2)$$

where A is the number of infected erythrocytes and B is the total erythrocyte count.

$$\%\text{suppression} = [A - B]/A \times 100 \qquad (3)$$

where A is the average percentage of parasitemia in the negative control group and B is the average percentage of parasitemia in the experimental groups.

## *In vivo* acute oral toxicity test

The acute oral toxicity of EMFL and EMFB was evaluated in the mouse models according to the standard guidelines of the Organization for Economic Cooperation and Development (OECD), test guideline no. 425 [43]. Briefly, 20 male ICR mice were randomly divided into four groups, each consisting of five mice per group. Group 1 served as the control group without any treatment; group 2 served as the negative control group and received the vehicle solvents (7% Tween 80 and 3% ethanol in distilled water); and groups 3 and 4 served as the experimental groups and received EMFL and EMFB at doses of 2,000 mg/kg body weight. Before beginning the experiments, the mice were fasted for 3 h. Following the fasting period, mice in groups 2, 3, and 4 received oral administration of the vehicle solvent and the EMFL and EMFB in a single dosage, administered in a volume of 200 μL *via* oral gavage, respectively. After treatment, gross physical and behavioral changes, such as rigidity, sleep, diarrhea, depression, abnormal secretion, and hair erection, were observed for a duration of 3 h and then monitored twice daily for a period of 14 days. Food and water consumption were recorded on a daily basis throughout the experimental periods. Additionally, the body weight

of the mice was measured before treatment on day 0 and upon completion of the experiment on day 14, using a sensitive digital weighing balance (Mettler Toledo, model: ML3002E, Indonesia). On day 14, all mice were anesthetized using 2% isoflurane (Piramal Critical Care, PA, USA). Blood samples were collected *via* cardiac puncture for biochemical analyses. Additionally, the liver and kidney were harvested, fixed with a 10% formalin solution, and stored at room temperature for 24–48 h. Subsequently, the specimens were used for the study of histopathological changes using hematoxylin and eosin (H&E) staining.

### Animal euthanasia procedure

At the end of the experiments, mice were euthanized by placing them in an anesthesia induction chamber to receive deep isoflurane anesthesia (Piramal Critical Care, PA, USA), which ensured unconsciousness and minimized pain and distress in accordance with established protocols [38]. Following the guidelines outlined by the American Veterinary Medical Association (AVMA) for the euthanasia of animals (2020 edition), a secondary method involving cervical dislocation was employed for sacrifice in order to confirm the death of animals.

### Analysis of liver and kidney function

Briefly, whole blood was centrifuged at 3,000 *g* for 5 min, and the plasma was collected for the analysis of biochemical parameters. Liver functions were evaluated by measuring levels of aspartate aminotransferase (AST), alanine aminotransferase (ALT), and alkaline phosphatase (ALP), while kidney functions were evaluated by measuring blood urea nitrogen (BUN) and creatinine levels, utilizing an AU480 chemistry analyzer (Beckman Coulter, Brea, CA, USA).

### Histopathological changes in the liver and kidney tissues

Histopathological changes in the liver and kidney tissues were investigated following standard histological processing with previously established protocols [27, 44, 45]. The fixed liver and kidney tissues in a 10% formalin solution underwent dehydration using a gradient series of ethanol solutions, clearing with xylene, and subsequent embedding in paraffin. The paraffin blocks were sectioned at a thickness of 5 μm, transferred to glass slides, and stained with H&E solution. The stained slides were then evaluated for histopathological changes under a light microscope (Olympus CX31, Tokyo, Japan) with a 40X objective lens. The assessment of histopathological changes was conducted independently by two different researchers, ensuring blindness for the experimental groups.

### Statistical analysis

The data presented in this study are expressed as the mean ± SEM (standard error of means). Statistical analyses were conducted using SPSS for Microsoft Windows version 17.0 (SPSS, IL, USA). A one-way analysis of variance (ANOVA) was conducted to assess the statistical significance of mean differences among groups, followed by a post hoc Turkey's multiple comparison test. This analysis was applied to various measured parameters, including percentage parasitemia, percentage parasitemia suppression, food consumption, water consumption, mean body weight, and liver and kidney function parameters. A *p*-value less than 0.05 ($p < 0.05$) was considered statistically significant.

**Table 1. Extraction yields of the EMFL, EMFB, AMFL, and AMFB.**

| Extracts | Initial weight of plant (g) | Weight of extract (g) | Yield (%, w/w) | Color and appearance |
|---|---|---|---|---|
| EMFL | 60.00 | 6.14 | 10.23% | Green sticky solid |
| EMFB | 60.00 | 1.41 | 2.35% | Brown sticky solid |
| AMFL | 60.00 | 4.58 | 7.63% | Dark brown solid |
| AMFB | 60.00 | 2.59 | 4.32% | Light brown solid |

EMFL: Ethanolic extract of *M. ferrea* L. leaves; EMFB: Ethanolic extract of *M. ferrea* L. branches; AMFL: Aqueous extract of *M. ferrea* L. leaves; AMFB: Aqueous extract of *M. ferrea* L. branches.

## Results

### Extraction yields of the plant extracts

The extraction yields of EMFL, EMFB, AMFL, and AMFB are summarized in Table 1. The EMFL provided the highest yield (6.14 g). The final product, a green sticky solid that resulted from freeze-drying, provided 10.23% yield with respect to the raw plant material used. Following closely, the AMFL weighed 4.58 g, or 7.63% yield, and exhibited a dark brown solid. Meanwhile, the AMFB weighed 2.59 g, or 4.32% yield, and exhibited a light brown solid. The EMFB provided the lowest yield of 2.35% (1.41 g), and the final product was a brown sticky solid.

### *In vitro* antimalarial activity of *M. ferrea* L. leaf and branch extracts

The antimalarial activity of the EMFL, EMFB, AMFL, and AMFB was evaluated using the pLDH assay against the *P. falciparum* K1 strain, as shown in Table 2. The classification of the antimalarial activity of the crude extracts was determined based on the $IC_{50}$ values, indicating their effectiveness in inhibiting the malaria parasite. Both the World Health Organization and previous studies on antimalarial drug discovery have defined plant extracts with $IC_{50}$ values <5 μg/mL as having high activity; $IC_{50}$ values ranging from 5 to 15 μg/mL are defined as

**Table 2. Antimalarial activity, cytotoxicity, and selectivity index (SI) values of the EMFL, EMFB, AMFL, and AMFB.**

| Samples | pLDH assay | MTT assay | SI values |
|---|---|---|---|
| | $IC_{50} \pm$ SEM (μg/mL) | $CC_{50} \pm$ SEM (μg/mL) | |
| | *P. falciparum* K1 | Vero cells | |
| Crude extracts | | | |
| EMFL | 6.76 ± 0.66 | 81.74 ± 4.87 | 12.09 |
| EMFB | 4.54 ± 0.46 | 46.16 ± 5.52 | 10.17 |
| AMFL | 54.50 ± 2.18 | >800 | >14.68 |
| AMFB | 46.23 ± 4.60 | 228.13 ± 4.90 | 4.93 |
| Compounds | | | |
| ARS* | 1.28 ± 0.03 | ND | ND |
| CQ* | 103.20 ± 4.50 | ND | ND |
| DX | ND | 1.82 ± 0.11 | ND |

The data are represented as the mean ± SEM.

* The data are expressed in ng/mL.

ND: Not determined.

$IC_{50}$: The half-maximal inhibitory concentration; $CC_{50}$: The 50% cytotoxic concentration; EMFL: Ethanolic extract of *M. ferrea* L. leaves; EMFB: Ethanolic extract of *M. ferrea* L. branches; AMFL: Aqueous extract of *M. ferrea* L. leaves; AMFB: Aqueous extract of *M. ferrea* L. branches; ARS: Artesunate; CQ: Chloroquine; DX: Doxorubicin.

having promising activity; IC$_{50}$ values ranging from 15 to 50 μg/mL are defined as having weak activity; and IC$_{50}$ values >50 μg/mL are defined as inactive [46]. In accordance with these classifications, among the four crude extracts, the EMFB exhibited high antimalarial activity with an IC$_{50}$ value of 4.54 ± 0.46 μg/mL. Following closely, the EMFL exhibited promising antimalarial activity with an IC$_{50}$ value of 6.76 ± 0.66 μg/mL. Conversely, AMFB and AMFL exhibited weak and inactive antimalarial activity, with IC$_{50}$ values of 46.23 ± 4.60 μg/mL and 54.50 ± 2.18 μg/mL, respectively.

### *In vitro* cytotoxicity and SI of *M. ferrea* L. leaf and branch extracts

The cytotoxicity of EMFL, EMFB, AMFL, and AMFB was evaluated using the MTT assay against Vero cells, as shown in Table 2. According to the criteria of the National Cancer Institute, plant extracts were considered to have cytotoxic effects if their CC$_{50}$ values were <30 μg/mL after 48–72 h of exposure to cells [47]. In this study, EMFB and EMFL exhibited non-toxicity to normal cells (Vero cells), with CC$_{50}$ values of 46.16 ± 5.52 and 81.74 ± 4.87 μg/mL, respectively. Similarly, AMFB and AMFL exhibited non-toxicity to Vero cells, with CC$_{50}$ values higher than those from the ethanolic extracts, reaching 228.13 ± 4.90 and >800 μg/mL, respectively. According to SI values, the AMFL exhibited the highest SI value of >14.68. Following closely, the EMFL and EMFB exhibited SI values of 12.09 and 10.17, respectively. However, the AMFB exhibited the lowest SI value of 4.93 (Table 2). The AMFL, as well as the EMFL and EMFB, met the acceptance criteria for SI values >10, indicating that these extracts were more selective against malarial parasites than normal cells. Despite passing the acceptance criteria, the AMFL demonstrated limited antimalarial activity, according to the IC$_{50}$ classification. Therefore, the EMFL and EMFB were chosen for further investigation of antimalarial activity and acute oral toxicity tests in mouse models due to their promising antimalarial activity, non-toxicity to normal cells, and meeting the acceptance criteria for SI values, respectively.

### GC-MS analysis of phytochemical constituents in EMFL and EMFB

The chromatograms that illustrate the GC-MS analysis of EMFL and EMFB are presented in Fig 1. The EMFL revealed the presence of 25 identified compounds, listed in Table 3, while the EMFB revealed 21 compounds, listed in Table 4. The most abundant compound in the EMFL was identified as friedelin, classified within the triterpenoid chemical compound subclass, with a retention time of 45.046 min. This compound accounted for 14.80% of the peak area, as depicted in Fig 1A. Following closely were friedelinol (6.44%), betulin (5.28%), β-caryophyllene (2.36%), oleamide (1.35%), and 5-hydroxymethylfurfural (1.08%), respectively. Among the total compounds, the most abundant compound subclass in the EMFL belonged to sesquiterpenoids, accounting for 32.00%. Noteworthy compounds within this subclass included α-copaene, cyperene, β-caryophyllene, α-humulene, alloaromadendrene, α-himachalene, β-cadinene, and β-caryophyllene oxide. Other compound subclasses comprised triterpenoids (12.00%), phenolic acids (12.00%), diterpenoids (8.00%), pyranones (8.00%), fatty acids (8.00%), aryl-aldehydes (4.00%), cinnamic acids (4.00%), vinylogous acids (4.00%), fatty acid esters (4.00%), and fatty acid amides (4.00%), respectively. The most abundant compound in the EMFB was identified as oleamide, classified within the fatty acid amide chemical compound subclass and eluting at a retention time of 28.407 min, which accounted for 4.88% of the peak area, as depicted in Fig 1B. Following closely were *cis*-β-farnesene (3.69%), alloaromadendrene (2.51%), physcion (1.59%), palmitic acid (1.36%), 5-hydroxymethylfurfural (1.09%), and 4H-pyran-4-one, 2,3-dihydro-3,5-dihydroxy-6-methyl- (1.04%), respectively. Among the total compounds, the most abundant compound subclass in the EMFB belonged to

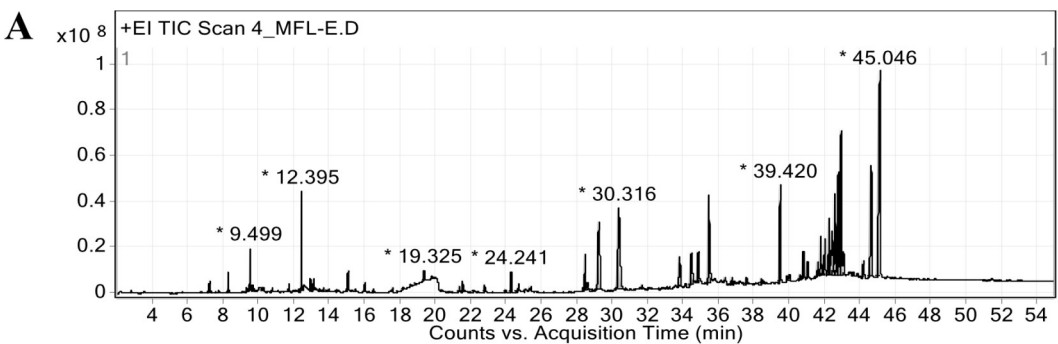

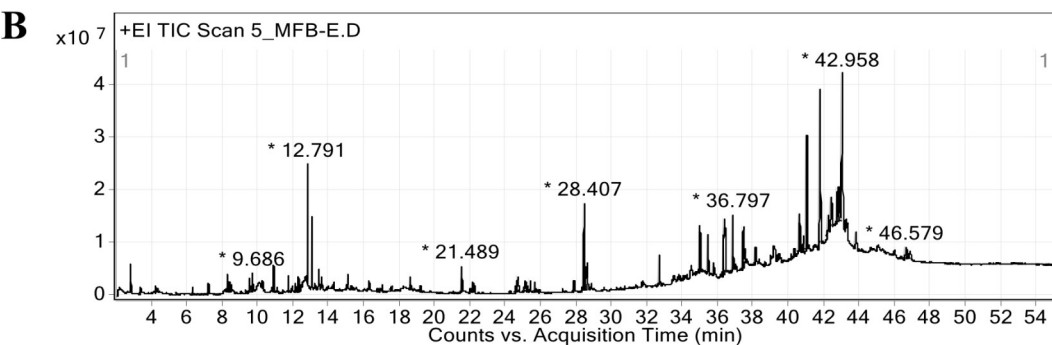

**Fig 1.** Gas chromatography-mass spectrometry chromatogram of the EMFL (A) and EMFB (B).

sesquiterpenoids, accounting for 33.33%. Compounds within this subclass included α-copaene, cyperene, β-caryophyllene, *cis*-β-farnesene, alloaromadendrene, α-selinene, and germacrene A. The other compound subclasses included fatty acids (9.52%), fatty acid esters (9.52%), fatty acid amides (9.52%), pyranones (4.76%), aryl-aldehydes (4.76%), phenolic acids (4.76%), cinnamic acid (4.76%), methoxyphenols (4.76%), hydrocarbons (4.76%), diterpenoids (4.76%), and anthraquinones (4.76%), respectively. Importantly, nine compounds were shared in both the EMFL and EMFB. These compounds included 4H-pyran-4-one, 2,3-dihydro-3,5-dihydroxy-6-methyl-, 5-hydroxymethylfurfural, α-copaene, cyperene, β-caryophyllene, alloaromadendrene, palmitic acid, ethyl palmitate, and oleamide.

## Four-day suppressive test results (Peter's test)

The percentages of parasitemia and suppression by the EMFL and EMFB are presented in Fig 2. Experimental groups of mice treated with both ethanolic extracts at all dosages exhibited a significant reduction in the percentage of parasitemia compared to the negative control group, where mice were treated with vehicle solvents ($p < 0.05$) (Fig 2A). The EMFL exhibited a significant suppression of *P. berghei* ANKA in a dose-dependent manner, with 41.15%, 47.96%, and 54.48% at doses of 200, 400, and 600 mg/kg body weight, respectively, compared to the negative control group ($p < 0.05$) (Fig 2B). Similarly, the EMFB also exhibited a significant suppression of *P. berghei* ANKA in a dose-dependent manner, with 35.43%, 42.17%, and 62.61%, respectively, compared to the negative control group ($p < 0.05$) (Fig 2B). Among the extract treatment groups, the EMFB at 600 mg/kg body weight exhibited the highest suppression of *P. berghei* ANKA with 62.61%. However, the standard antimalarial drug treatment groups, including 6 mg/kg body weight of ARS and 25 mg/kg body weight of CQ, exhibited the highest significant suppression of *P. berghei* ANKA with 95.63% and 97.99%, respectively, when compared to all treatment groups ($p < 0.05$) (Fig 2B).

**Table 3. Identified phytochemical constituents in the EMFL by GC-MS.**

| No | RT (min) | Compound name | Chemical subclass | Molecular formula | Molecular weight | Peak area (%) |
|---|---|---|---|---|---|---|
| 1 | 45.046 | Friedelin | Triterpenoids | $C_{30}H_{50}O$ | 426 | 14.80 |
| 2 | 44.555 | Friedelinol | Triterpenoids | $C_{30}H_{52}O$ | 428 | 6.44 |
| 3 | 42.697 | Betulin | Triterpenoids | $C_{30}H_{50}O_2$ | 442 | 5.28 |
| 4 | 12.395 | β-Caryophyllene* | Sesquiterpenoids | $C_{15}H_{24}$ | 204 | 2.36 |
| 5 | 28.410 | Oleamide* | Fatty acid amides (primary amides) | $C_{18}H_{35}NO$ | 281 | 1.35 |
| 6 | 9.499 | 5-Hydroxymethylfurfural* | Aryl-aldehydes | $C_6H_6O_3$ | 126 | 1.08 |
| 7 | 24.241 | Phytol | Diterpenoids | $C_{20}H_{40}O$ | 296 | 0.63 |
| 8 | 15.025 | β-Caryophyllene oxide | Sesquiterpenoids | $C_{15}H_{24}O$ | 220 | 0.55 |
| 9 | 13.110 | 3-Hydroxybenzoic acid | Phenolic acids | $C_7H_6O_3$ | 138 | 0.48 |
| 10 | 8.270 | 4H-Pyran-4-one, 2,3-dihydro-3,5-dihydroxy-6-methyl-* | Pyranones | $C_6H_8O_4$ | 144 | 0.45 |
| 11 | 21.495 | Palmitic acid* | Fatty acids (saturated) | $C_{16}H_{32}O_2$ | 256 | 0.42 |
| 12 | 7.225 | Maltol | Pyranones | $C_6H_6O_3$ | 126 | 0.41 |
| 13 | 24.661 | Linolenic acid | Fatty acids (unsaturated) | $C_{18}H_{30}O_2$ | 278 | 0.36 |
| 14 | 19.325 | Phytol acetate | Diterpenoids | $C_{22}H_{42}O_2$ | 338 | 0.32 |
| 15 | 22.745 | Hulupone | Vinylogous acids | $C_{20}H_{28}O_4$ | 332 | 0.32 |
| 16 | 15.969 | Homovanillic acid | Phenolic acids | $C_9H_{10}O_4$ | 182 | 0.28 |
| 17 | 12.907 | α-Humulene | Sesquiterpenoids | $C_{15}H_{24}$ | 204 | 0.27 |
| 18 | 13.024 | Alloaromadendrene* | Sesquiterpenoids | $C_{15}H_{24}$ | 204 | 0.20 |
| 19 | 11.718 | α-Copaene* | Sesquiterpenoids | $C_{15}H_{24}$ | 204 | 0.18 |
| 20 | 17.538 | Coniferyl alcohol | Phenols | $C_{10}H_{12}O_3$ | 180 | 0.15 |
| 21 | 13.658 | α-Himachalene | Sesquiterpenoids | $C_{15}H_{24}$ | 204 | 0.09 |
| 22 | 13.943 | β-Cadinene | Sesquiterpenoids | $C_{15}H_{24}$ | 204 | 0.07 |
| 23 | 12.117 | Cyperene* | Sesquiterpenoids | $C_{15}H_{24}$ | 204 | 0.06 |
| 24 | 19.769 | Shikimic acid | Phenolic acids | $C_7H_{10}O_5$ | 174 | 0.06 |
| 25 | 22.129 | Ethyl palmitate* | Fatty acid esters (wax monoesters) | $C_{18}H_{36}O_2$ | 284 | 0.06 |

RT: Retention time

*Identical phytochemical constituents were found in both ethanolic extracts.

### *In vivo* acute oral toxicity test results

The effects of EMFL and EMFB were evaluated in an acute oral toxicity test in mouse models. This evaluation involved the oral administration of a single dosage of both ethanolic extracts at 2,000 mg/kg body weight. Both groups of mice treated with ethanolic extracts showed no signs of acute toxicity within the first 24 h and throughout the experiments. There were no changes in physical and behavioral signs such as rigidity, sleep, diarrhea, depression, abnormal secretion, and hair erection when compared to both mice in the control group and the negative control group, which received vehicle solvents. Moreover, no mortality was observed in both groups of mice treated with ethanolic extracts throughout the 14-day experiment period. Regarding the effects of both ethanolic extracts on food and water consumption, both groups of mice treated with ethanolic extracts exhibited no significant difference in the average food and water consumption in weeks 1 and 2 compared to the control and negative control groups ($p > 0.05$) (Fig 3A and 3B). In addition, both groups of mice treated with ethanolic extracts exhibited no significant difference in the mean of body weight changes compared to the control and negative control groups ($p > 0.05$) (Fig 4). As a result, the median lethal dose ($LD_{50}$) for both EMFL and EMFB is estimated to be greater than 2,000 mg/kg body weight.

**Table 4. Identified phytochemical constituents in the EMFB by GC-MS.**

| No | RT (min) | Compound name | Chemical subclass | Molecular formula | Molecular weight | Peak area (%) |
|---|---|---|---|---|---|---|
| 1 | 28.407 | Oleamide* | Fatty acid amides (primary amides) | $C_{18}H_{35}NO$ | 281 | 4.88 |
| 2 | 12.791 | *cis*-β-Farnesene | Sesquiterpenoids | $C_{15}H_{24}$ | 204 | 3.69 |
| 3 | 13.024 | Alloaromadendrene* | Sesquiterpenoids | $C_{15}H_{24}$ | 204 | 2.51 |
| 4 | 32.649 | Physcion | Anthraquinones | $C_{16}H_{12}O_5$ | 284 | 1.59 |
| 5 | 21.489 | Palmitic acid* | Fatty acids (saturated) | $C_{16}H_{32}O_2$ | 256 | 1.36 |
| 6 | 9.499 | 5-Hydroxymethylfurfural* | Aryl-aldehydes | $C_6H_6O_3$ | 126 | 1.09 |
| 7 | 8.267 | 4H-Pyran-4-one, 2,3-dihydro-3,5-dihydroxy-6-methyl-* | Pyranones | $C_6H_8O_4$ | 144 | 1.04 |
| 8 | 12.276 | Cinnamic acid | Phenolic acids | $C_9H_8O_2$ | 148 | 0.81 |
| 9 | 22.124 | Ethyl palmitate* | Fatty acid esters (wax monoesters) | $C_{18}H_{36}O_2$ | 284 | 0.65 |
| 10 | 25.363 | Palmitamide | Fatty acid amides (primary amides) | $C_{16}H_{33}NO$ | 255 | 0.61 |
| 11 | 11.718 | α-Copaene* | Sesquiterpenoids | $C_{15}H_{24}$ | 204 | 0.50 |
| 12 | 8.497 | Benzoic acid | Phenolic acids | $C_7H_6O_2$ | 122 | 0.49 |
| 13 | 25.612 | Geranylgeraniol | Diterpenoids | $C_{20}H_{34}O$ | 290 | 0.49 |
| 14 | 13.597 | Germacrene A | Sesquiterpenoids | $C_{15}H_{24}$ | 204 | 0.45 |
| 15 | 24.545 | Linoleic acid | Fatty acids (unsaturated) | $C_{18}H_{32}O_2$ | 280 | 0.39 |
| 16 | 22.219 | Eicosane | Hydrocarbons | $C_{20}H_{42}$ | 282 | 0.37 |
| 17 | 25.195 | Ethyl Oleate | Fatty acid esters (wax diesters) | $C_{20}H_{38}O_2$ | 310 | 0.33 |
| 18 | 12.117 | Cyperene* | Sesquiterpenoids | $C_{15}H_{24}$ | 204 | 0.29 |
| 19 | 12.392 | β-Caryophyllene* | Sesquiterpenoids | $C_{15}H_{24}$ | 204 | 0.29 |
| 20 | 13.548 | α-Selinene | Sesquiterpenoids | $C_{15}H_{24}$ | 204 | 0.29 |
| 21 | 16.901 | Methoxyeugenol | Methoxyphenols | $C_{11}H_{14}O_3$ | 194 | 0.16 |

RT: Retention time

* Identical phytochemical constituents were found in both ethanolic extracts.

## Liver and kidney function analysis results

The effects of EMFL and EMFB on plasma biochemical parameters associated with liver and kidney functions were evaluated (Fig 5). Regarding the biochemical parameters associated with liver functions, mice treated with 2,000 mg/kg body weight of the EMFL exhibited a significant increase in ALT levels compared to the control and negative control groups ($p < 0.05$) (Fig 5B). Meanwhile, the AST and ALP levels exhibited no significant difference compared to the control and negative control groups ($p > 0.05$) (Fig 5A and 5C). In contrast, mice treated with 2,000 mg/kg body weight of the EMFB exhibited no significant difference in AST, ALT, and ALP levels compared to those from both the control and negative control groups ($p > 0.05$) (Fig 5A–5C). Biochemical parameters associated with kidney functions were also investigated. Both groups of mice treated with ethanolic extracts exhibited no significant differences in BUN and creatinine levels compared to the control and negative control groups ($p > 0.05$) (Fig 5D and 5E).

## Histopathological changes observed in liver and kidney tissues

The effects of EMFL and EMFB on histopathological changes in liver and kidney tissues were investigated (Fig 6). The histopathological examinations of liver tissues from mice treated with 2,000 mg/kg body weight of EMFL (Fig 6E) and EMFB (Fig 6G) revealed a normal architecture, with a normal appearance of the central vein and hepatic sinusoid. The hepatocytes

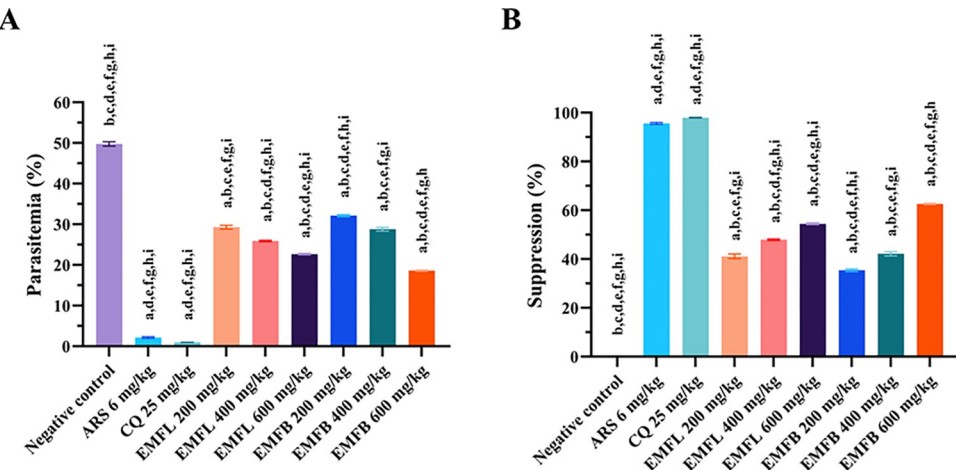

**Fig 2.** The effects of EMFL and EMFB on percentage parasitemia (A) and suppression (B) in *P. berghei* ANKA-infected mice in a 4-day suppressive test. The data are represented as the mean ± SEM (*n* = 5 per group). [a] compared to the negative control group receiving vehicle solvent (7% Tween 80 and 3% ethanol in distilled water); [b] compared to the positive control group receiving ARS at 6 mg/kg body weight; [c] compared to the positive control group receiving CQ at 25 mg/kg body weight; [d] compared to the experimental group receiving EMFL at 200 mg/kg body weight; [e] compared to the experimental group receiving EMFL at 400 mg/kg body weight; [f] compared to the experimental group receiving EMFL at 600 mg/kg body weight; [g] compared to the experimental group receiving EMFB at 200 mg/kg body weight; [h] compared to the experimental group receiving EMFB at 400 mg/kg body weight; [i] compared to the experimental group receiving EMFB at 600 mg/kg body weight. *p* < 0.05 was considered statistically significant.

appeared normal, with polygonal shapes and a spherical and centralized nucleus. This observation was consistent with the findings from the mice in the control group (Fig 6A) and the negative control group (Fig 6C). Furthermore, no congestion in the central vein or inflammatory cellular infiltrations were observed in liver tissues in all groups. Similarly, the kidney tissues from mice treated with 2,000 mg/kg body weight of EMFL (Fig 6F) and EMFB (Fig 6H) revealed intact glomeruli, normal renal tubular structures, a fine Bowman's capsule, and Bowman's space. This observation was consistent with the findings from the mice in the control group (Fig 6B) and the negative control group (Fig 6D). Additionally, no congestion or inflammatory cellular infiltration in the interstitial was observed in all groups. Based on this finding, histopathological examinations of the liver and kidney tissues of mice treated with both EMFL and EMFB revealed no anatomical abnormalities.

## Discussion

The increasing levels of *P. falciparum* resistance to artemisinin derivatives and their partner drugs pose an urgent threat to global malaria control and eradication efforts [1, 3]. Therefore, there is an urgent need to identify novel and potent antimalarial drugs [8]. Medicinal plants have long been recognized as a source of molecules with therapeutic potential and an extensive storehouse of ethnobotanical compounds [9, 48]. Histologically, the successful antimalarial drugs that have been derived from medicinal plants provide hope that these plants may serve as sources for the development of further potent antimalarial drugs [12]. This study focuses on *M. ferrea* L., a plant traditionally used for a variety of purposes, with antipyretic, antimicrobial, carminative, expectorant, cardiotonic, and diuretic properties, as well as treatment of colds and asthma [19]. Extensive studies have mostly focused on the antimalarial activity of *M. ferrea* L. flowers [21, 24, 34, 40]. However, there is no scientific evidence claiming antimalarial activity in other parts of *M. ferrea* L., such as its leaves and branches. As a result, the current

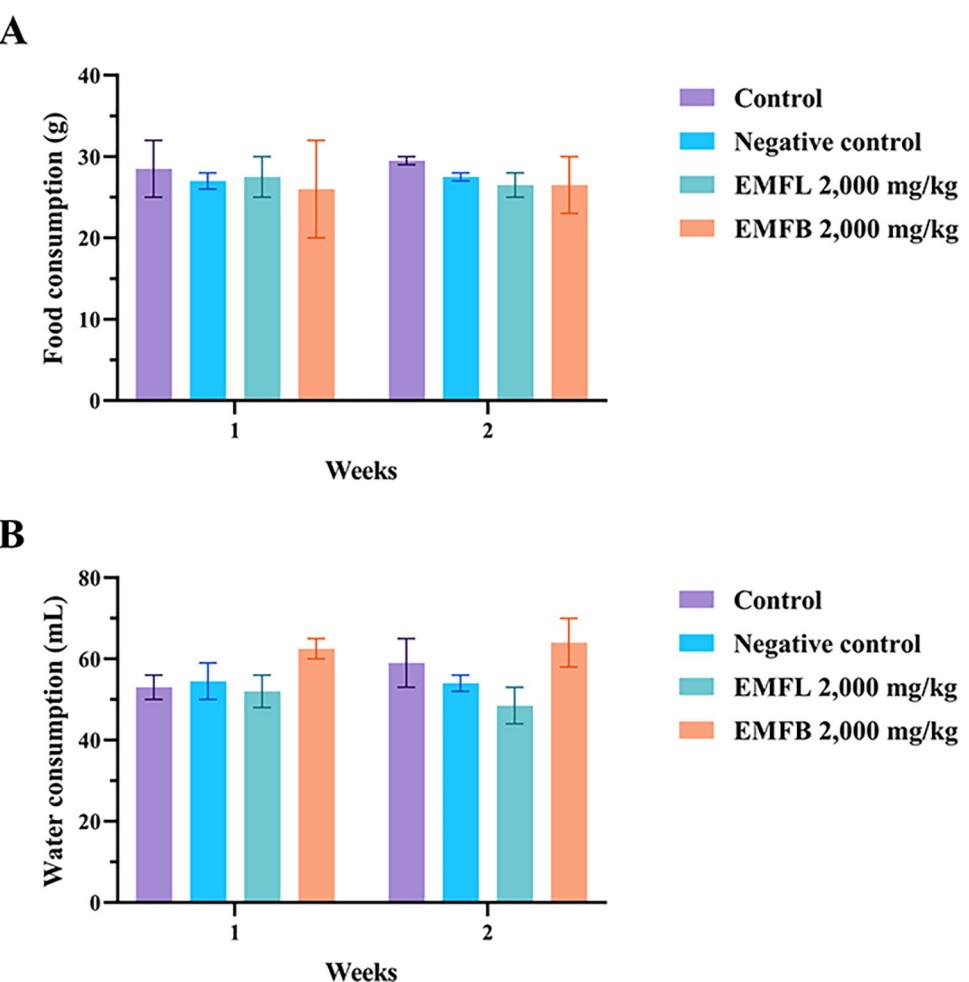

**Fig 3.** The effects of EMFL and EMFB on food (A) and water consumption (B) in an acute oral toxicity test. The data are represented as the mean ± SEM (*n* = 5 per group).

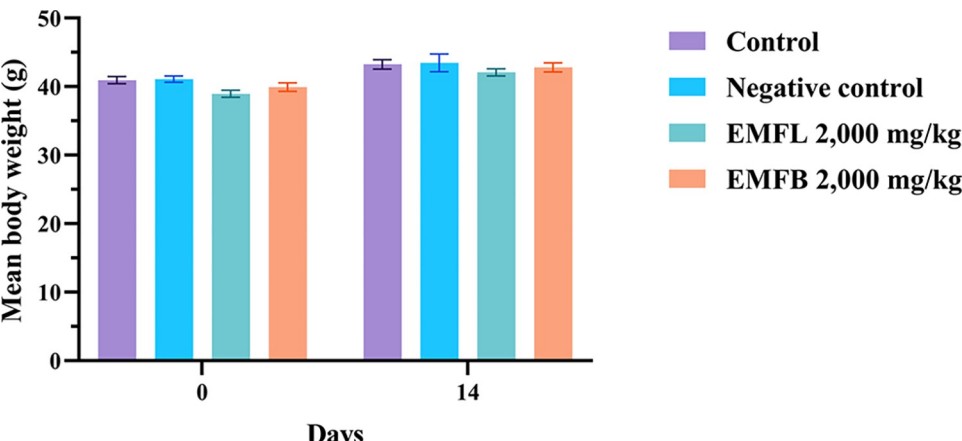

**Fig 4. The effects of EMFL and EMFB on mean body weight changes in an acute oral toxicity test.** The mean body weight change was measured before treatment (day 0) and after the completion of the experiments (day 14). The data are represented as the mean ± SEM (*n* = 5 per group).

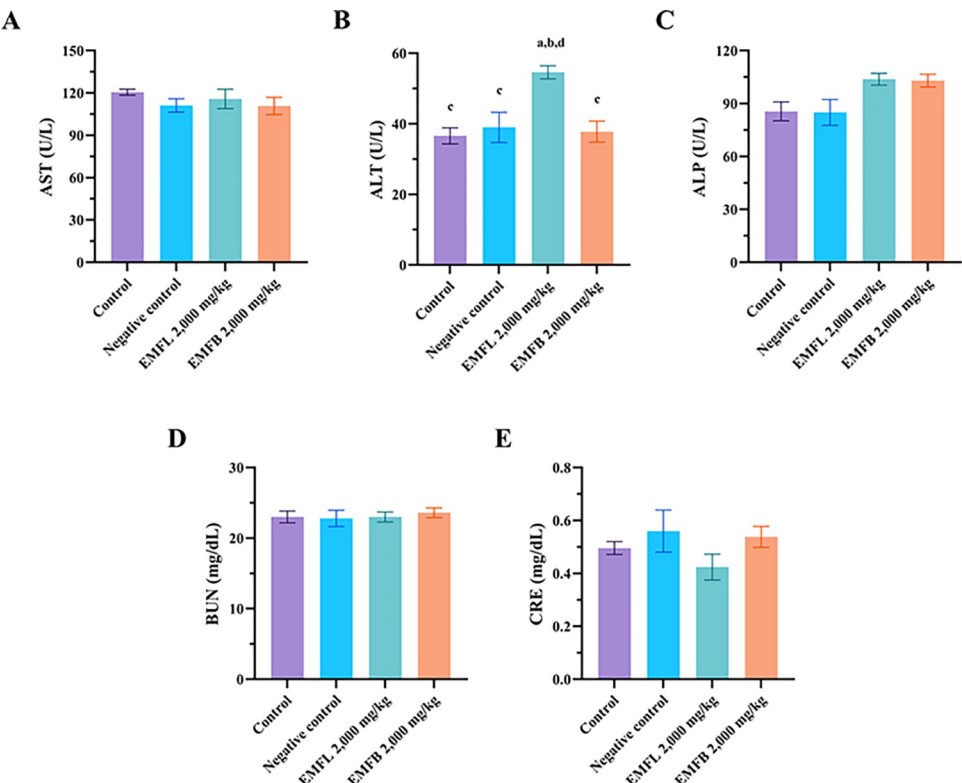

**Fig 5. The effects of EMFL and EMFB on liver and kidney functions in an acute oral toxicity test.** The liver function parameters measured include aspartate aminotransferase (AST; A), alanine aminotransferase (ALT; B), and alkaline phosphatase (ALP; C), while kidney function parameters include blood urea nitrogen (BUN; D) and creatinine (CRE; E). The data are represented as the mean ± SEM (*n* = 5 per group). [a] compared to the control group; [b] compared to the negative control group receiving vehicle solvent (7% Tween 80 and 3% ethanol in distilled water); [c] compared to the experimental group receiving EMFL at 2,000 mg/kg body weight; [d] compared to the experimental group receiving EMFB at 2,000 mg/kg body weight. $p < 0.05$ is considered statistically significant.

investigation aimed to address existing gaps in scientific research by evaluating the antimalarial activity and safety profiles of *M. ferrea* L. leaves and branches.

In this investigation, the *in vitro* antimalarial activity of crude extracts derived from *M. ferrea* L. leaves and branches was evaluated using the pLDH assay against the *P. falciparum* K1 strain. Among the four crude extracts, the EMFB exhibited high antimalarial activity (IC$_{50}$ = 4.54 ± 0.46 μg/mL), and the EMFL exhibited promising antimalarial activity (IC$_{50}$ = 6.76 ± 0.66 μg/mL). In contrast, the AMFB and AMFL exhibited weak and inactive antimalarial activity, respectively. A higher antimalarial activity was observed in the ethanolic extracts when compared with the aqueous extracts because aqueous solvents are unable to extract lipophilic phytochemicals that are efficiently extracted by ethanol [49, 50]. Therefore, it is conceivable that these specific phytochemicals may be responsible for the observed antimalarial activity. Previous studies have demonstrated that ethanolic extracts derived from *M. ferrea* L. leaves have significantly stronger antioxidant activity compared with other extracts, primarily due to their higher phenolic and alkaloid content [25]. Furthermore, ethanolic extracts derived from the *M. ferrea* L. stem bark have good antioxidant activities, providing over 90% protection to erythrocytes, hemoglobin, and DNA, which is known to be associated with the higher phenolic and total flavonoid content in the extracts [51]. In addition, methanolic extracts derived from the *M. ferrea* L. stem bark exhibit potent anti-inflammatory activity, inhibiting

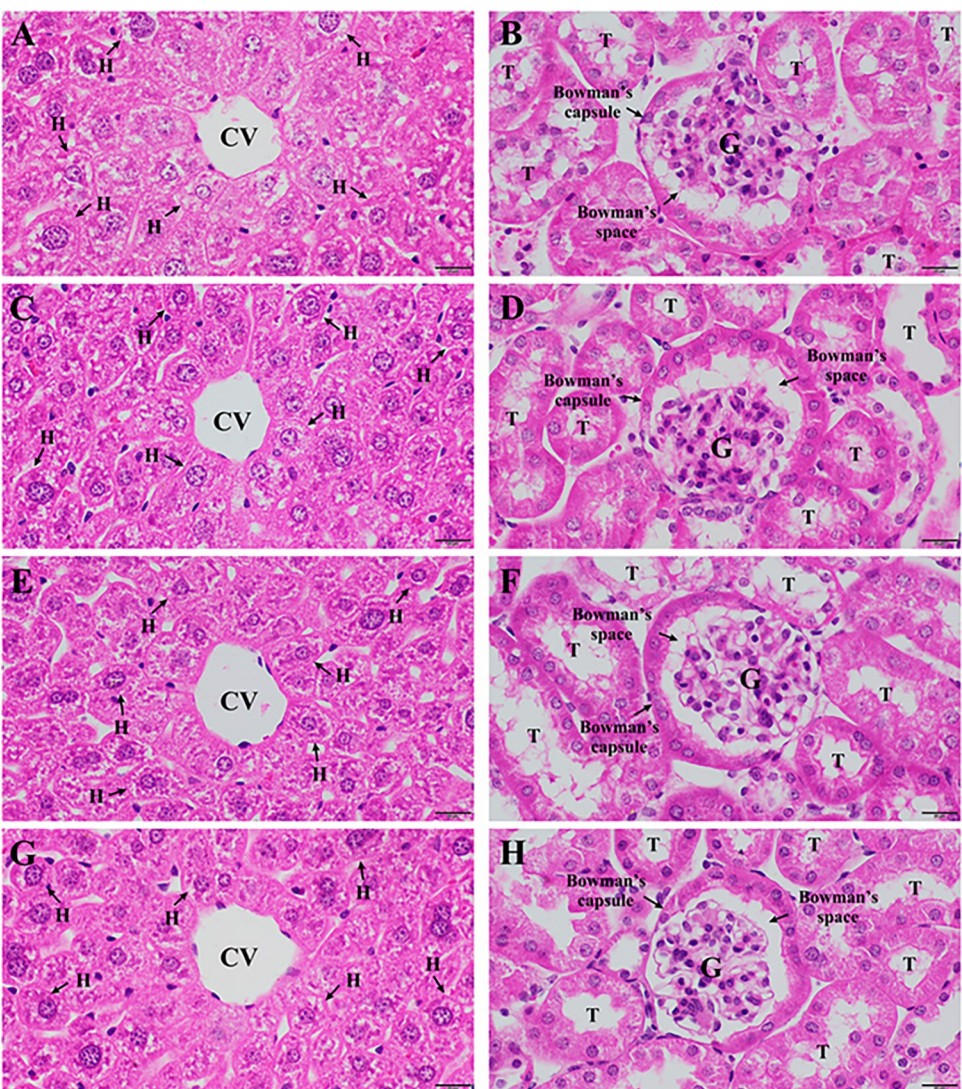

**Fig 6. Histopathological examinations of the liver and kidney tissues in an acute oral toxicity test.** Control group (A, B). Negative control group (7% Tween 80 and 3% ethanol in distilled water) (C, D). Experimental group (2,000 mg/kg body weight of EMFL) (E, F). Experimental group (2,000 mg/kg body weight of EMFB) (G, H). All the tissues were stained with hematoxylin and eosin. All images were photographed at 400x magnification with a scale bar = 20 μm. Central vein (CV), hepatocytes (H), glomerulus (G), and renal tubules (T).

nitric oxide production in the murine macrophage RAW 264.7 cell line with an $IC_{50}$ of 63.36 μg/mL [52]. Previous studies have investigated the antimalarial activity of *Mesua* coumarins isolated from the blossoms of *M. ferrea* L. using the pLDH assay. These compounds exhibited antimalarial activity, with $IC_{50}$ values ranging from 1.17 to 13.75 μg/mL against *P. falciparum* W2 strain [21]. Additionally, isoprenylated coumarins derived from the *Mesua borneensis* L. stem bark exhibited antimalarial activity, with an $IC_{50}$ value ranging from 1.02 to 8.81 μg/mL against the chloroquine-sensitive strain *P. falciparum* 3D7 strain [53].

The *in vitro* cytotoxicity of the four crude extracts was evaluated using the MTT assay against Vero cells. All crude extracts were shown to be non-toxic to Vero cells, with $CC_{50}$ values >30 μg/mL. The crude extracts with higher SI values, particularly those >10, are regarded as more promising due to their pronounced selectivity against malaria parasites [32, 33, 54].

Among the four crude extracts, both the EMFL and EMFB met the acceptance criteria with SI values >10. In addition, they exhibited promising antimalarial activity based on the $IC_{50}$ value classifications and were also non-toxic to Vero cells. Therefore, both ethanolic extracts were selected for potency assessment in subsequent *in vivo* antimalarial activity and acute oral toxicity studies in mouse models.

*In vivo* proof-of-concept studies for promising candidates from *in vitro* antimalarial studies typically involve the use of rodent animal models [55]. These models have been employed to identify several standard antimalarial drugs, including mefloquine, halofantrine, and artemisinin derivatives [56]. The 4-day suppressive test (Peter's test) is the most widely used method for evaluating the capability of compounds to suppress RMP in the early stages of antimalarial drug discovery programs [55, 56]. RMP are frequently employed in experimental *in vivo* studies owing to their shared basic biology, biochemical processes, conserved genome organization, and the pattern of sensitivity and resistance to drugs with the human malaria parasite [57]. The RMP, *P. berghei* ANKA, is a lethal strain known for inducing severe pathological effects in rodents, such as cerebral malaria [58]. This strain also serves as a well-characterized model for understanding the pathogenesis of human cerebral malaria [59, 60]. Mice infected with *P. berghei* ANKA experience rapid disease progression, resulting in ataxia, paralysis, and coma, with eventual mortality due to brain hemorrhages, sequestration of infected erythrocytes in the cerebral vasculature, accumulation of various inflammatory cells, and edema [61, 62]. In addition, mice infected with *P. berghei* are commonly employed for evaluating the efficacy of most antimalarial drug candidates in clinical development, such as KAF156, KAE609, OZ349, and DDD107498 [55].

In the current study, the 4-day suppressive test was employed to evaluate the efficacy of both EMFL and EMFB to suppress the RMP, *P. berghei* ANKA, in ICR mouse models. The results demonstrated that all groups of mice treated with ethanolic extracts at all dosages exhibited a higher capacity to suppress *P. berghei* ANKA-infected mice in a dose-dependent manner, compared to the negative control group, which received the vehicle solvents ($p < 0.05$). Among all extract-treated groups, the highest percentage of suppression was observed in the maximal dosage of 600 mg/kg body weight, i.e., 54.48% and 62.61% for EMFL and EMFB, respectively, compared to the negative control group ($p < 0.05$). However, it is noteworthy that the percentage of parasitemia suppression in all groups of mice treated with ethanolic extracts showed a significantly lower efficacy compared to the groups treated with antimalarial drugs, specifically ARS (95.63%) and CQ (97.99%). The lower parasitemia suppression observed with both EMFL and EMFB in comparison to ARS and CQ may be attributed to the lower concentration of active compounds present in the extracts. However, compounds exhibiting a parasitemia suppression rate ≥ 30% compared to the negative control group were considered active against malaria infection [63–65]. In accordance with these criteria, both ethanolic extracts, at all dosages, exhibited a parasitemia suppression rate > 30% and were thus presumed to be active. In the context of the *in vivo* antimalarial activity of *M. ferrea* L. extracts, the ethanolic extract derived from *M. ferrea* L. flowers has been chosen as one of the ingredients in a novel formulation (CPF-1). CPF-1 demonstrated the ability to achieve a parasitemia suppression rate of 72.01% in the 4-day suppressive test at the maximal dosage of 600 mg/kg body weight [34]. In addition, *M. ferrea* L. flowers are an ingredient in the PSCD remedy, a Thai traditional medicine used to treat fever. The PSCD remedy exhibited a parasitemia suppression rate of 78.36% in the 4-day suppressive test at the maximal dosage of 600 mg/kg body weight [24]. In light of these findings, one potential application could involve combining both EMFL and EMFB with other medicinal plants to enhance antimalarial efficacy and potentially mitigate toxicity associated with other medicinal plant ingredients. However, further experimental studies must be conducted to validate all of these claims.

The acute toxicity test aims to determine the short-term adverse effects of a single high dosage of the test substances. This test, which is typically performed during the early stages of new substance studies, provides critical information on the potential toxicity of the substance under investigation [66]. The hematopoietic system serves as a sensitive target for toxic substances, providing an important indicator of physiological and pathological conditions in both humans and animals [67]. Furthermore, hematological parameters play a critical role in toxicity studies when extrapolating experimental data into clinical trials [68]. The liver and kidney serve as primary organs in drug metabolism and excretion, since they are frequently exposed to toxic substances [68, 69]. Regarding hepatocellular function, AST, ALT, and ALP are recognized as liver marker enzymes that exhibit increased activity during liver injury [70, 71]. More specifically, the increased levels of AST and ALT are attributed to the toxic-induced transformation of cellular membrane permeability, resulting in the release of these enzymes into the systematic circulation [66, 70]. Conversely, the levels of creatinine and BUN have long been considered the "gold standard" for identifying drug-induced nephrotoxicity and renal failure [72].

In the current study, both groups of mice treated with 2,000 mg/kg of EMFL and EMFB showed no signs of toxicity, and no mortality occurred during the observed period of the experiment. In addition, when compared to the control and negative control groups, mice treated with an EMFB did not elevate the plasma biochemical parameters associated with liver and kidney functions, which included the AST, ALT, ALP, BUN, and creatinine levels ($p > 0.05$). These findings are consistent with the histopathological examination results of liver and kidney tissues, both of which demonstrated a normal morphology. Similarly, mice treated with an EMFL did not elevate their AST, ALP, BUN, and creatinine levels compared to the control and negative control groups ($p > 0.05$). Conversely, there was a significant increase in ALT levels compared to the control and negative control groups ($p < 0.05$). Moreover, the histopathological examination of liver tissue confirmed the absence of structural changes in the integrity of liver cells. According to preclinical study guidelines, a two- to four-fold increase or higher in ALT activity alone in individual or group mean data, compared with concurrent controls, should be considered a potential indicator of hepatocellular injury [73, 74]. In this study, the ALT levels in mice treated with the EMFL exhibited an increase of less than twofold (1.49 times) compared to the control group, suggesting that the EMFL did not induce toxic effects on liver tissue and liver functions. These findings align with those of previous studies, which reported that oral administration of methanolic extracts of *M. ferrea* L. leaves did not result in mortality or acute adverse effects in Swiss albino mouse models [75, 76]. In addition, changes in body weight were used to evaluate the possible side effects of drugs and chemicals [67]. Both groups of mice treated with ethanolic extracts exhibited no significant difference in body weight changes compared to the control and negative control groups ($p > 0.05$). Moreover, all groups of mice exhibited weight gain throughout the experiments. A possible reason for the finding is that both groups of mice treated with ethanolic extracts showed no negative impact on appetite or water or food consumption, allowing for normal growth in the mice. Our findings are supported by the fact that both groups of mice treated with ethanolic extracts exhibited no significant differences in food and water consumption behavior compared to the control and negative control groups ($p > 0.05$). Therefore, the $LD_{50}$ value for both ethanolic extracts was greater than 2,000 mg/kg body weight when administered orally in a single dose. According to the Globally Harmonized System of Classification and Labelling of Chemicals, both ethanolic extracts were classified as category 5 (minimum toxic substances) [43]. However, it is recommended that both ethanolic extracts should be further investigated using sub-acute toxicity tests to establish extensive safety profiles before advancing to human clinical trials.

GC-MS is one of the most effective, rapid, and accurate techniques to identify a wide range of phytochemical constituents, such as alcohols, alkaloids, nitro compounds, long-chain hydrocarbons, organic acids, steroids, esters, and amino acids [77]. This technique combines GC to separate the phytochemical constituents of plant extracts and MS to analyze each constituent separately [78]. The GC-MS triple quadrupole becomes crucial in the search for bioactive compounds and the conduct of chemotaxonomic investigations on medicinal plants containing biologically active components [79]. Consequently, this study employed the GC-MS technique for the investigation of phytochemical constituents in EMFL and EMFB.

The GC-MS analysis revealed that 25 and 21 phytochemical constituents were identified in the EMFL and EMFB, respectively. The most abundant compound identified in *M. ferrea* L. leaves was friedelin, a triterpenoid compound. Friedelin has been recognized to possess various biological properties, including anti-inflammatory, antibacterial, and antiviral properties [80]. Friedelin, isolated from the *Psorospermum glaberrimum* stem bark, shows significant antimalarial activity against *P. falciparum* W2 strain, with an $IC_{50}$ value of 7.70 μM [81]. A similar finding was observed with friedelin isolated from *Endodesmia calophylloides*, which demonstrated antimalarial activity against *P. falciparum* W2 strain, with an $IC_{50}$ value of 7.2 μM [82]. Additionally, the triterpenoid compound friedelinol, a reduced derivative of friedelin, was identified in *M. ferrea* L. leaves. Prior investigations have reported that friedelinol, also isolated from the *Vismia laurentii* stem bark, possesses no activity against *P. falciparum* W2 strain [83]. Furthermore, betulin, another triterpenoid compound, was identified in *M. ferrea* L. leaves. Betulin and its derivatives have been widely studied for various biological properties, including antiviral, anti-cancer, antimalarial, and anti-inflammatory properties [84]. Previous studies reported that betulin, which is isolated from the *Ampelozizyphus amazonicus* root bark, exhibited significant antimalarial activity against both chloroquine-sensitive *P. falciparum* 3D7 strain and chloroquine-resistant *P. falciparum* Dd2 strain, with $IC_{50}$ values of 17.08 and 14.22 μM, respectively [85]. Additionally, another study reported that betulin exhibited antiplasmodial activity with an $IC_{50}$ value of $< 27$ μM, and its possible mechanism of action involves forming hydrogen bonds with the erythrocyte membrane, leading to structural changes that prohibit merozoite invasion and growth [86]. Moreover, other compound subclasses, such as diterpenoids and fatty acids, were identified in *M. ferrea* L. leaves. Phytol, a diterpenoid, has been found particularly in *M. ferrea* L. leaves. Previous studies found that phytol, a diterpenoid compound isolated from *Cassia fistula* leaves, exhibited antimalarial activity against chloroquine-sensitive *P. falciparum* D10 strain, with an $IC_{50}$ value of 18.90 μM [87]. Furthermore, phytol has been demonstrated to suppress *P. berghei* by 24.63% in a 4-day suppressive test at a dose of 20 mg/kg body weight. Histological analyses indicate that phytol also provides protective effects against the destruction of brain cells caused by *P. berghei* infection [88]. Furthermore, linolenic acid, a fatty acid compound, was also identified in *M. ferrea* L. leaves. Previous studies have reported that linolenic acid, isolated from *Calea tenuifolia* leaves, demonstrated antimalarial activity against chloroquine-sensitive *P. falciparum* PoW strain and chloroquine-resistant *P. falciparum* Dd2 strain, with $IC_{50}$ values of 49.6 and 142.0 μM, respectively [89]. The most abundant compound identified in *M. ferrea* L. branches was oleamide, a fatty acid amide. Oleamide exhibits a wide range of biological activities, including sleep induction, immunological suppression, and the activation of serotonin and GABA receptors [90]. Interestingly, previous studies have revealed that oleamide, a component of the *Guatteria recurvisepala* extract, has a 3.5-fold increase in relative binding affinity to *Plasmodium falciparum* thioredoxin reductase. This increased binding affinity is consistent with *in vitro* antimalarial activity against *P. falciparum* K1 strain, with an $IC_{50}$ value of 4.29 μg/mL [91]. Furthermore, various compound subclasses, such as methoxyphenols, diterpenoids, fatty acids, and anthraquinones, were identified in *M. ferrea* L. branches. Methoxyeugenol, a

methoxyphenol compound, was identified in *M. ferrea* L. branches. Studies have demonstrated that methoxyeugenol, found in the *Myristica fragrans* extract, exhibits diverse biological activities, including anti-helminthic, antimicrobial, anti-inflammatory, and anticancer properties [92]. In addition, the derivative eugenol has been reported to significantly reduce parasitemia and prevent both cerebral edema and cognitive dysfunction in *P. berghei* ANKA infection [93]. Geranylgeraniol, a diterpenoid compound, was also identified in *M. ferrea* L. branches. Previous investigations have reported that this compound exhibits good antimalarial activity, particularly against *P. falciparum* K1 strain, with an $IC_{50}$ value of 1.07 μg/mL [94]. Additionally, linoleic acid, a fatty acid compound, was found in *M. ferrea* L. branches. Studies have shown that linoleic acid, isolated from *C. tenuifolia* leaves, exhibits antimalarial activity against *P. falciparum* PoW strain and *P. falciparum* Dd2 strain, with $IC_{50}$ values of 21.8 and 31.1 μM, respectively [89]. Furthermore, linoleic acid has been reported to mitigate inflammatory responses by reducing the protein expression of tumor necrosis factor-α and interleukin-1β in pheochromocytoma cells (PC12 cell line) [95]. Moreover, physcion, an anthraquinone compound, has been identified in *M. ferrea* L. branches. Previous studies have demonstrated that physcion, isolated from fungal endophytes associated with *A. annua* L., demonstrated potent antimalarial activity against *P. falciparum* NF54 strain, with an $IC_{50}$ value of 0.9 μM [96]. Anthraquinones, as a subclass, are known for their diverse pharmacological properties, including purgative properties, anti-inflammatory, immunoregulation, anti-hyperlipidemia, and anticancer effects [97]. Additionally, anthraquinones have been reported to intercalate parasite DNA due to their cyclic planar structure [98].

Interestingly, our analysis revealed that nine compounds were shared between *M. ferrea* L. leaves and branches. Previous investigations have highlighted that the GC-MS analysis of essential oils derived from *M. ferrea* L. bark, tender leaves, and mature leaves, revealed the presence of certain shared compounds. Specifically, α-copaene and β-caryophyllene were identified as constituents common to these different plant parts [99]. Among nine compounds, β-caryophyllene is a sesquiterpenoid compound. Studies have highlighted the antimalarial potential of β-caryophyllene, as evidenced by its activity against *P. falciparum* 3D7 strain, with an $IC_{50}$ of 8.25 μg/mL, as well as its *in vivo* effectiveness against the chloroquine-sensitive strain *P. berghei* NK65, resulting in a remarkable 88.2% suppression of parasitemia at a dosage of 100 mg/kg body weight [100]. Furthermore, both leaves and branches were found to contain palmitic acid, a fatty acid. Prior investigations have reported that palmitic acid was found to be the main component in all extracts, including those from *Fadogiella stigmatoloba*, *Hygrophila auriculata*, *Hylodesmum repandum*, and *Porphyrostemma chevalieri*. Significantly, all extracts exhibited both antimalarial and antioxidant activities [101]. Additionally, ethyl palmitate, another identified compound, is recognized for its diverse biological activities, including hypocholesterolemic, antioxidant, nematicidal, and antimicrobial properties [102].

This study highlighted the abundance of two major compound classes in both *M. ferrea* L. leaves and branches: terpenoids (including sesquiterpenoids, diterpenoids, and triterpenoids) and fatty acids (including fatty acids, fatty acid esters, and fatty acid amides). The leaves accounted for 52.00% of terpenoids and 16.00% of fatty acids, while the branches accounted for 38.09% of terpenoids and 28.56% of fatty acids. Terpenoids, particularly those derived from medicinal plants, such as sesquiterpenoids (C15), diterpenoids (C20), and triterpenoids (C30), have been recognized for their various biological properties, including anti-cancer, anti-inflammatory, antibacterial, antiviral, and antimalarial properties [103]. Moreover, fatty acids are also known to possess a wide range of biological activities, including antibacterial and antifungal properties [104]. Prior investigations have reported that essential oils isolated from *Hexalobus crispiflorus* exhibited antimalarial activity against *P. falciparum* W2 strain, with an $IC_{50}$ value of 2.0 μg/mL. These essential oils contain a high sesquiterpene content of

99.5%. Hence, the observed antimalarial effects are probably related to the high concentration of sesquiterpenoids [105]. Furthermore, previous studies have revealed that the oleoresin derived from *Copaifera reticulata* exhibited significant *in vivo* antimalarial activity against *P. berghei* ANKA, achieving a remarkable 93% suppression at a dosage of 100 mg/kg/day. These observed activities are likely attributed to the presence of sesquiterpenoids, with β-caryophyllene (41.7%) and β-bisabolene (18.6%) identified as the predominant constituents in the oleoresin [106]. Furthermore, it has been reported that fatty acids can exhibit antimalarial activity. Prior investigations demonstrated that very long chain *cis* $C_{23}$–$C_{26}$ Δ5,9 fatty acids derived from the sponge *Agelas oroides* exhibited antimalarial activity against *P. falciparum* K1 strain, with an $IC_{50}$ value ranging from 12 to 16 μg/mL. This antimalarial activity has been linked with significant inhibition against *P. falciparum* enoyl-ACP reductase, the final reduction step in the fatty acid chain elongation cycle in *P. falciparum* [107]. Additionally, other studies have indicated that polyunsaturated fatty acids have greater antimalarial activity when compared to saturated and monounsaturated fatty acids [57, 108].

Based on the GC-MS analysis of the EMFL and EMFB, it is evident that these extracts contain a high amount of terpenoids and fatty acids, accounting for more than 60% of the total phytochemical constituents. Consequently, the observed antimalarial activity in this study may be attributed to the abundant presence of terpenoids and fatty acids in the extracts. Importantly, these ethanolic extracts also exhibit a diverse range of phytochemical constituents, implying that various components may contribute to the observed antimalarial activity through different mechanisms of action. For instance, physcion may exert its effects by intercalating with parasite DNA, while oleamide may interact with the enzyme *P. falciparum* thioredoxin reductase. Additionally, triterpenoids such as betulin may interact with the erythrocyte membrane, thereby inhibiting merozoite invasion and growth. The primary goal is to inhibit malaria parasites, as supported by the aforementioned phytochemical constituents. Moreover, these mechanisms of phytochemical constituents may operate either individually or synergistically to contribute to antimalarial activity in the observed study.

## Conclusions

This study is the first to demonstrate *in vitro* antimalarial activity against the *P. falciparum* K1 strain and *in vivo* antimalarial activity against *P. berghei* ANKA using EMFL and EMFB. Both ethanolic extracts contain various phytochemical constituents that may exert effects against malaria parasites, either individually or synergistically. Additionally, neither of the ethanolic extracts induced signs of acute toxicity in the mouse models when administered orally at a single dosage of 2,000 mg/kg body weight. However, to validate these promising findings, further experimental investigations are warranted for both ethanolic extracts. This includes evaluating subacute toxicity tests to establish extensive safety profiles before considering them as alternative treatments for malaria in humans. Furthermore, it is crucial to isolate the bioactive compounds in the EMFL and EMFB, investigate their antimalarial potential, and identify their targets in malaria parasite cells. This endeavor is critical for understanding their mechanisms of action, which could serve as a scaffold for the development of potent and novel antimalarial drugs.

## Supporting information

**S1 File. Data tables for graph generation.**
(PDF)

**S2 File. Histopathological examinations of the liver and kidney tissues in an acute oral toxicity test (uncropped raw images).**
(PDF)

## Acknowledgments

The authors express gratitude to the Center for Scientific and Technological Equipment of Walailak University for their valuable support and assistance in providing laboratory equipment and facilities for this study. Additionally, we express appreciation to the staff at the Department of Tropical Pathology, Faculty of Tropical Medicine, Mahidol University, Thailand, for their assistance with histological preparations and staining. We also give special thanks to Miss Rungruedee Kimseng from the Research Institute for Health Sciences (RIHS), Walailak University, for her support in the animal experiments.

## Author Contributions

**Conceptualization:** Atthaphon Konyanee, Prapaporn Chaniad, Chuchard Punsawad.

**Data curation:** Atthaphon Konyanee, Prapaporn Chaniad, Chuchard Punsawad.

**Formal analysis:** Atthaphon Konyanee, Prapaporn Chaniad, Arisara Phuwajaroanpong, Walaiporn Plirat, Parnpen Viriyavejakul, Abdi Wira Septama, Chuchard Punsawad.

**Funding acquisition:** Atthaphon Konyanee, Chuchard Punsawad.

**Investigation:** Atthaphon Konyanee, Prapaporn Chaniad, Arisara Phuwajaroanpong, Walaiporn Plirat, Chuchard Punsawad.

**Methodology:** Atthaphon Konyanee, Prapaporn Chaniad, Arisara Phuwajaroanpong, Walaiporn Plirat, Parnpen Viriyavejakul, Abdi Wira Septama, Chuchard Punsawad.

**Project administration:** Prapaporn Chaniad, Chuchard Punsawad.

**Resources:** Prapaporn Chaniad, Parnpen Viriyavejakul, Chuchard Punsawad.

**Validation:** Atthaphon Konyanee, Prapaporn Chaniad, Chuchard Punsawad.

**Writing – original draft:** Atthaphon Konyanee, Prapaporn Chaniad, Chuchard Punsawad.

**Writing – review & editing:** Atthaphon Konyanee, Prapaporn Chaniad, Parnpen Viriyavejakul, Abdi Wira Septama, Chuchard Punsawad.

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
