## [Decision Letter · Decision Letter 0]

24 Jul 2024

PONE-D-24-18759Exploring the potential antimalarial properties, safety profile, and phytochemical composition of Mesua ferrea Linn.PLOS ONE

Dear Dr. Punsawad,

Thank you for submitting your manuscript to PLOS ONE. After careful consideration, we feel that it has merit but does not fully meet PLOS ONE’s publication criteria as it currently stands. Therefore, we invite you to submit a revised version of the manuscript that addresses the points raised during the review process.

We look forward to receiving your revised manuscript.

Kind regards,

Yash Gupta, Ph.D.

Academic Editor

PLOS ONE

2. To comply with PLOS ONE submissions requirements, in your Methods section, please provide additional information regarding the experiments involving animals and ensure you have included details on methods of sacrifice.

Additional Editor Comments:

This paper has unclear methodology and needs an overall elaboration to meet the publication quality.

Please also clarify the following:

1. How was the sample size of the animal experiments determined and which statistical test were used.

2. State the name of the Institutional Animal Care and Use Committee (IACUC) or other

relevant ethics board that reviewed the study protocol and formally accepted the study. As malaria infection model is a post procedure survival strategy the ethical clearance cannot be waived off.

3. Standard guidelines of the Organization for Economic Cooperation and Development does not amount to ethical clearance.

Reviewers' comments:

Reviewer's Responses to Questions

**Comments to the Author**

1. Is the manuscript technically sound, and do the data support the conclusions?

Reviewer #1: Yes

Reviewer #2: Yes

2. Has the statistical analysis been performed appropriately and rigorously? 

Reviewer #1: Yes

Reviewer #2: Yes

3. Have the authors made all data underlying the findings in their manuscript fully available?

Reviewer #1: Yes

Reviewer #2: Yes

4. Is the manuscript presented in an intelligible fashion and written in standard English?

Reviewer #1: Yes

Reviewer #2: Yes

5. Review Comments to the Author

Reviewer #1: Exploring the potential antimalarial properties, safety profile, and phytochemical composition of Mesua ferrea Linn.

This study has highlighted the importance of phytochemicals to combat malarial parasite. The study investigated the therapeutic potential of Mesua ferrea Linn., against Plasmodium falciparum’s resistance to lead drug resistance to global malaria control. In vitro assays evaluated with crude extracts from M. ferrea leaves and branches shows impressive data (4.54 µg/mL), followed by the ethanolic extract of leaves. Besides the study has been replicated in mouse model holds its promising antimalarial activity. Overall, this study highlights the potential of M. ferrea as a natural antimalarial resource

However, these are some aspects which authors should address:

1)In the material and methods section the human ethical clearance is stated. However, no such data has been presented in this study. It is unnecessary and should be removed.

2)All the data have been presented in form of table. These can be made into graphical representation. This would increase the readability of the article.

3)The arrows in the histological section should be made bold.

4)Further, Plasmodium berghei anka is a model for celebral malaria. Authors may discuss the effect of extracts on the brain inflammation.

5)The peak area represents the abundance of the compound in the extract. Compounds may be listed in the order of their abundance

The study shows promising results and is well written with relevant references.

Reviewer #2: Atthaphon Konyanee et al ‘Exploring the potential antimalarial properties, safety profile, and phytochemical composition of Mesua ferrea Linn. Here are some constructive comments and observations you might consider for the provided results:

1.Line 148-149 It's important to explicitly state whether written or verbal informed consent was obtained from participants. Please clarify this detail.

2.Could you provide more information on the anesthesia protocols and criteria for humane endpoints?

3.Could you specify if any permits or ethical considerations were required for the collection of M. ferrea L. leaves and branches?

4.Can you provide more details on the statistical methods used to analyze the data, particularly in terms of the ANOVA model and post hoc tests applied? Additionally, were power analyses conducted to determine the sample sizes?

5.It would strengthen the manuscript to include more explicit details on the protocols used, including any modifications from previously established methods. This would enhance transparency and facilitate replication of the study.

6.In the acute oral toxicity test, how were the criteria for assessing toxicity determined, and what were the specific endpoints used for determining humane euthanasia?

7.Given the detailed methodology for plant extraction, it would be beneficial to include any information regarding potential conflicts of interest in the selection and processing of M. ferrea L. leaves and branches.

8.Are there studies or hypotheses regarding the mechanism by which the ethanolic extracts exert their antimalarial activity? How do these mechanisms relate to the identified phytochemicals?

9.Please provide a clear Figure 2 with an arrow indicating the designated paraphrase area.

6. PLOS authors have the option to publish the peer review history of their article (what does this mean?). If published, this will include your full peer review and any attached files.

Reviewer #1: No

Reviewer #2: **Yes: **Manish Shukla

---

## [Author Response · Author response to Decision Letter 0]

1 Sep 2024

Point-by-point responses to the reviewers’ comments

Additional Editor Comments:

This paper has unclear methodology and needs an overall elaboration to meet the publication quality.

Please also clarify the following:

1. How was the sample size of the animal experiments determined and which statistical test were used.

Response: In this study, we employed G*Power software version 3.1.9.6 [1] to calculate the required sample sizes for the in vivo antimalarial activity evaluated through a 4-day suppressive test. The statistical analysis conducted in G*Power software employed an “ANOVA: fixed effect, omnibus, one-way analysis” with the type of power analysis set to “A priori: compute required sample size - given �, power, and effect size” in accordance with previous studies [2]. The effect size was set at 0.8, the � error prob at 0.05, and the power (1�� error prob) at 0.95, indicating a statistical power of 95%. With nine groups defined, this resulted in a minimum required sample size of forty-five experimental mice, allocated as five mice per group, as indicated in the manuscript; please see pages 14�15 and lines 309�319. For the acute oral toxicity test, we adhered to the Organization for Economic Cooperation and Development (OECD), test guideline no. 425 [3], which recommends using five mice per group for limit tests at a dosage of 2,000 mg/kg body weight. Consequently, we employed four groups of mice for this test, resulting in a total of twenty experimental mice, as indicated in the manuscript; please see pages 16 and lines 345�349. The sample size calculations and the use of animals in this study were approved by the Walailak University Institutional Animal Care and Use Committee (WU-IACUC) under the approval number WU-ACUC-66019.

References:

1. Faul F, Erdfelder E, Buchner A, Lang A-G. Statistical power analyses using G*Power 3.1: Tests for correlation and regression analyses. Behav Res Methods. 2009;41(4):1149-60. doi: 10.3758/BRM.41.4.1149.

2. Amadi PU, Agomuo EN, Ukaga CN, Njoku UC, Amadi JA, Nwaekpe CG. Preclinical trial of traditional plant remedies for the treatment of complications of gestational Malaria. Medicines (Basel). 2021;8(12). Epub 20211217. doi: 10.3390/medicines8120079. PubMed PMID: 34940291; PubMed Central PMCID: PMCPMC8703497.

3. OECD. Test No. 425: Acute Oral Toxicity: Up-and-Down Procedure2022.

2. State the name of the Institutional Animal Care and Use Committee (IACUC) or other

relevant ethics board that reviewed the study protocol and formally accepted the study. As malaria infection model is a post procedure survival strategy the ethical clearance cannot be waived off.

Response: This study employed mouse models for both the 4-day suppressive test and the acute oral toxicity test. Prior to conducting the animal experiments, the study protocol was submitted to the Animal Ethics Committee at Walailak University. After a comprehensive review, the animal ethical clearance was obtained from the Walailak University Institutional Animal Care and Use Committee (WU-IACUC), with the approval number WU-ACUC-66019. All procedures were conducted in accordance with relevant guidelines and regulations for the use of animals, in compliance with the Animal Research: Reporting of In Vivo Experiments (ARRIVE) guidelines. Furthermore, the information relevant to animal use has been included in the PLOS ONE Humane Endpoints Checklist and the ARRIVE Guidelines 2.0: Author Checklist, which were submitted alongside the manuscript. For information, please see pages 7 and lines 152�159.

3. Standard guidelines of the Organization for Economic Cooperation and Development does not amount to ethical clearance.

Response: The acute oral toxicity test is essential for determining a safer dose range to manage the clinical signs and symptoms associated with drug administration [4]. To minimize the number of animals required for testing and to reduce animal stress, the OECD has established test guidelines, including test guideline no. 425 [3, 5], which was accepted as an alternative to the median lethal dose (LD50) method in 2008 and revised in 2022 [6]. This guideline is widely recognized as a standard protocol for acute oral toxicity testing and is accepted by U.S. and international regulatory authorities, including the European Chemical Agency (ECHA), as an alternative method for chemical safety testing [7, 8]. In this study, we submitted study protocols relevant to the acute oral toxicity tests, adhering to OECD guidelines, to the Animal Ethics Committee at Walailak University. Following a thorough review, we obtained animal ethical clearance from the Walailak University Institutional Animal Care and Use Committee (WU-IACUC), with the approval number WU-ACUC-66019. Additionally, our literature review indicates that several studies have successfully utilized OECD guidelines for acute oral toxicity tests of various test substances [4, 9-11]. For information on animal ethics and the OECD guidelines, please see pages 7 (lines 152�159) and pages 16 (lines 343-345).

References:

3. OECD. Test No. 425: Acute Oral Toxicity: Up-and-Down Procedure2022.

4. Saleem U, Amin S, Ahmad B, Azeem H, Anwar F, Mary S. Acute oral toxicity evaluation of aqueous ethanolic extract of Saccharum munja Roxb. roots in albino mice as per OECD 425 TG. Toxicol Rep. 2017;4:580-5. Epub 20171031. doi: 10.1016/j.toxrep.2017.10.005. PubMed PMID: 29152463; PubMed Central PMCID: PMCPMC5671618.

5. Kojima H, Nakada T, Yagami A, Todo H, Nishimura J, Yagi M, et al. A step-by-step approach for assessing acute oral toxicity without animal testing for additives of quasi-drugs and cosmetic ingredients. Curr Res Toxicol. 2023;4:100100. Epub 20221223. doi: 10.1016/j.crtox.2022.100100. PubMed PMID: 36619289; PubMed Central PMCID: PMCPMC9816657.

6. Sewell F, Ragan I, Horgan G, Andrew D, Holmes T, Manou I, et al. New supporting data to guide the use of evident toxicity in acute oral toxicity studies (OECD TG 420). Regul Toxicol Pharmacol. 2024;146:105517. Epub 20231012. doi: 10.1016/j.yrtph.2023.105517. PubMed PMID: 37838350.

7. Alternative methods accepted by US agencies. Available from: https://ntp.niehs.nih.gov/whatwestudy/niceatm/accept-methods.

8. Alternative methods for toxicity testing. Available from: https://joint-research-centre.ec.europa.eu/reference-measurement/european-union-reference-laboratories/eu-reference-laboratory-alternatives-animal-testing-eurl-ecvam/alternative-methods-toxicity-testing/validated-test-methods-health-effects/acute-toxicity_en.

9. Rocha JMV, de Souza VB, Panunto PC, Nicolosi JS, da Silva EDN, Cadore S, et al. In vitro and in vivo acute toxicity of a novel citrate-coated magnetite nanoparticle. PLoS One. 2022;17(11):e0277396. Epub 20221117. doi: 10.1371/journal.pone.0277396. PubMed PMID: 36395271; PubMed Central PMCID: PMCPMC9671459.

10. Uppu DS, Manjunath GB, Yarlagadda V, Kaviyil JE, Ravikumar R, Paramanandham K, et al. Membrane-active macromolecules resensitize NDM-1 gram-negative clinical isolates to tetracycline antibiotics. PLoS One. 2015;10(3):e0119422. Epub 20150319. doi: 10.1371/journal.pone.0119422. PubMed PMID: 25789871; PubMed Central PMCID: PMCPMC4366164.

11. Mulatu A, Megersa N, Tolcha T, Alemu T, Vetukuri RR. Antifungal compounds, GC-MS analysis and toxicity assessment of methanolic extracts of Trichoderma species in an animal model. PLoS One. 2022;17(9):e0274062. Epub 20220923. doi: 10.1371/journal.pone.0274062. PubMed PMID: 36149851; PubMed Central PMCID: PMCPMC9506656.

Reviewer #1: Exploring the potential antimalarial properties, safety profile, and phytochemical composition of Mesua ferrea Linn.

This study has highlighted the importance of phytochemicals to combat malarial parasite. The study investigated the therapeutic potential of Mesua ferrea Linn., against Plasmodium falciparum’s resistance to lead drug resistance to global malaria control. In vitro assays evaluated with crude extracts from M. ferrea leaves and branches shows impressive data (4.54 µg/mL), followed by the ethanolic extract of leaves. Besides the study has been replicated in mouse model holds its promising antimalarial activity. Overall, this study highlights the potential of M. ferrea as a natural antimalarial resource.

However, these are some aspects which authors should address:

1. In the material and methods section the human ethical clearance is stated. However, no such data has been presented in this study. It is unnecessary and should be removed.

Response: In this study, we utilized human blood samples from participants. Erythrocytes were isolated from blood samples and subsequently used for the in vitro cultivation of the Plasmodium falciparum K1 strain to evaluate the antimalarial activity through the parasite lactate dehydrogenase (pLDH) assay, as detailed on pages 9 and lines 188�190. Furthermore, we have included a statement regarding the ethical clearance for the use of human blood samples in the ethics approval section; please see pages 7 and lines 144�150.

2. All the data have been presented in form of table. These can be made into graphical representation. This would increase the readability of the article.

Response: Thank you very much for your valuable comments. In response to your recommendations, we have made revisions to improve the clarity and readability of our manuscript. Specifically, we have transformed tables 5, 6, 7, and 8 into figures 2, 3, 4, and 5, respectively, to enhance the visual representation of our findings and facilitate easier comprehension. Please see pages 26�27 (lines 504�516), pages 28 (lines 538�553), pages 29 (lines 560�570), and pages 30�31 (lines 585�595) respectively. However, upon further evaluation, we concluded that tables 1, 2, 3, and 4 were not suitable for conversion into graphical format. Therefore, we have retained these tables in their original format to maintain optimal readability and ensure that the information is easily understood by the reader.

3. The arrows in the histological section should be made bold.

Response: Thank you very much for your valuable comments. I have improved figure 6 by enlarging and bolding the arrows, letter symbols, and descriptions related to the histopathological examination of the liver and kidney tissues in order to enhance readability. Please see figure 6, pages 32�33 and lines 615�622.

4. Further, Plasmodium berghei ANKA is a model for cerebral malaria. Authors may discuss the effect of extracts on the brain inflammation.

Response: Thank you very much for your valuable comments. In response to your recommendations, I have included a discussion of several compounds reported to exhibit protective effects on the brain during P. berghei ANKA infections, specifically those identified in the ethanolic extracts of M. ferrea L. leaves and branches (EMFL and EMFB). For instance, the diterpenoid phytol present in EMFL has been shown to have protective effects against brain damage induced by P. berghei infection [12]. Additionally, eugenol, a derivative of methoxyeugenol present in EMFB, has been reported to prevent cerebral edema and cognitive dysfunction associated with the P. berghei ANKA infection [13]. Please see discussion section on pages 40 (lines 792�795) and pages 41 (lines 810�812). Furthermore, several compounds and subclasses identified in EMFL and EMFB demonstrate anti-inflammatory activity, as discussed in the relevant section. These include friedelin, betulin, methoxyeugenol, linoleic acid, anthraquinone, and terpenoids. The presence of these compounds in the extracts may contribute to the mitigation of the inflammatory response during P. berghei ANKA infections, either individually or through synergistic interactions.

References:

12. Usman MA, Usman FI, Abubakar MS, Salman AA, Adamu A, Ibrahim MA. Phytol suppresses parasitemia and ameliorates anaemia and oxidative brain damage in mice infected with Plasmodium berghei. Exp Parasitol. 2021;224:108097. Epub 2021/03/20. doi: 10.1016/j.exppara.2021.108097. PubMed PMID: 33736972.

13. Pontes KAO, Silva LS, Santos EC, Pinheiro AS, Teixeira DE, Peruchetti DB, et al. Eugenol disrupts Plasmodium falciparum intracellular development during the erythrocytic cycle and protects against cerebral malaria. Biochim Biophys Acta Gen Subj. 2021;1865(3):129813. Epub 20201213. doi: 10.1016/j.bbagen.2020.129813. PubMed PMID: 33321150.

5. The peak area represents the abundance of the compound in the extract. Compounds may be listed in the order of their abundance

Response: Thank you very much for your valuable comments. In response, I have rearranged the order of the compounds identified in the ethanolic extracts of M. ferrea L. leaves and branches (EMFL and EMFB) through gas chromatography-mass spectrometry (GC-MS) analysis. The compounds are now presented in descending order based on their abundance, as determined by peak area. Please see tables 3 (pages 23�24, lines 481�483) and 4 (pages 24�25, lines 485�487).

6. The study shows promising results and is well written with relevant references.

Response: Thank you very much.

Reviewer #2: Atthaphon Konyanee et al ‘Exploring the potential antimalarial properties, safety profile, and phytochemical composition of Mesua ferrea Linn. Here are some constructive comments and observations you might consider for the provided results:

1. Line 148-149 It’s important to explicitly state whether written or verbal informed consent was obtained from participants. Please clarify this detail.

Response: In this study, we utilized human blood samples from participants for the in vitro cultivation of the human malaria P. falciparum K1 strain. Prior to blood sample collection, our study protocol was approved by the Ethics Committee in Human Research Walailak University (approval number: WUEC-23-062-01). We also provided participants with an information sheet outlining the details of the study, along with an informed consent form. After reviewing this information, participants who agreed to take part in the study provided written informed consent. Blood samples were then collected by an expert medical technologist, as described in the ethics approval and consent participation section. Please see pages 7 and lines 144�150.

2. Could you provide more information on the anesthesia protocols and criteria for humane endpoints?

Response: In this study, we employed the inhalation method to induce anesthesia in mice using 2% isoflurane (Piramal Critical Care, PA, USA) mixed with oxygen as the carrier gas within an anesthesia induction chamber, following an established protocol [14]. This approach aimed to minimize potential distress in the mice. To verify successful anesthesia induction, we assessed the mice by evaluating their toe-pinch pain reflex, following an established protocol [15]. Additionally, the animals were routinely monitored twice daily, and any animals displaying signs of coma or experiencing severe symptoms, including immobility, lack of body extension, and unresponsiveness to external stimuli, were immediately euthanized in a humane manner to alleviate pain and distress [16]. Euthanasia was performed under deep isoflurane anesthesia, in accordance with the established protocol [16]. Please see pages 13�14 (lines 292�297), pages 14 (lines 301�305), pages 15 (lines 329�331), and pages 16 (lines 358�359), respectively. Moreover, we have also included this information in the PLOS ONE Humane Endpoints Checklist document.

References:

14. Ebert G, Lopaticki S, O'Neill MT, Steel RWJ, Doerflinger M, Rajasekaran P, et al. Targeting the extrinsic pathway of hepatocyte apoptosis promotes clearance of Plasmodium liver infection. Cell Rep. 2020;30(13):4343-54.e4. doi: 10.1016/j.celrep.2020.03.032. PubMed PMID: 32234472.

15. Sarfo BY, Armah HB, Irune I, Adjei AA, Olver CS, Singh S, et al. Plasmodium yoelii 17XL infection up-regulates RANTES, CCR1, CCR3 and CCR5 expression, and induces ultrastructural changes in the cerebellum. Malar J. 2005;4:63. Epub 20051216. doi: 10.1186/1475-2875-4-63. PubMed PMID: 16359553; PubMed Central PMCID: PMCPMC1343570.

16. Matsubara R, Aonuma H, Kojima M, Tahara M, Andrabi SB, Sakakibara H, et al. Plant hormone salicylic acid produced by a malaria parasite controls host immunity and cerebral malaria outcome. PLoS One. 2015;10(10):e0140559.

---

## [Decision Letter · Decision Letter 1]

30 Sep 2024

Exploring the potential antimalarial properties, safety profile, and phytochemical composition of Mesua ferrea Linn.

PONE-D-24-18759R1

Dear Dr. Punsawad,

We’re pleased to inform you that your manuscript has been judged scientifically suitable for publication and will be formally accepted for publication once it meets all outstanding technical requirements.

Kind regards,

Yash Gupta, Ph.D.

Academic Editor

PLOS ONE

Additional Editor Comments (optional):

Reviewers' comments:

Reviewer's Responses to Questions

**Comments to the Author**

1. If the authors have adequately addressed your comments raised in a previous round of review and you feel that this manuscript is now acceptable for publication, you may indicate that here to bypass the “Comments to the Author” section, enter your conflict of interest statement in the “Confidential to Editor” section, and submit your "Accept" recommendation.

Reviewer #1: All comments have been addressed

Reviewer #2: All comments have been addressed

2. Is the manuscript technically sound, and do the data support the conclusions?

Reviewer #1: Yes

Reviewer #2: Yes

3. Has the statistical analysis been performed appropriately and rigorously? 

Reviewer #1: Yes

Reviewer #2: Yes

4. Have the authors made all data underlying the findings in their manuscript fully available?

Reviewer #1: Yes

Reviewer #2: Yes

5. Is the manuscript presented in an intelligible fashion and written in standard English?

Reviewer #1: Yes

Reviewer #2: Yes

6. Review Comments to the Author

Reviewer #1: Authors have responded to all query raised.They have also included the table and figures which makes the article more readable and is suited for publication.

Reviewer #2: reviewed the revised paper on the potential antimalarial properties, safety profile, and phytochemical composition of Mesua ferrea Linn., and I found that all comments have been adequately addressed.

7. PLOS authors have the option to publish the peer review history of their article (what does this mean?). If published, this will include your full peer review and any attached files.

Reviewer #1: No

Reviewer #2: No

---

## [Editor Report · Acceptance letter]

19 Nov 2024

PONE-D-24-18759R1 

PLOS ONE

Dear Dr. Punsawad, 

I'm pleased to inform you that your manuscript has been deemed suitable for publication in PLOS ONE. Congratulations! Your manuscript is now being handed over to our production team.

Kind regards, 

on behalf of

Dr. Yash Gupta 

Academic Editor

PLOS ONE